# Site-specific glycosylation regulates the form and function of the intermediate filament cytoskeleton

Heather J Tarbet[1], Lee Dolat[2,3], Timothy J Smith[1], Brett M Condon[1], E Timothy O'Brien III[1,4], Raphael H Valdivia[2,3], Michael Boyce[1,3]*

[1]Department of Biochemistry, Duke University School of Medicine, Durham, United States; [2]Department of Molecular Genetics and Microbiology, Duke University School of Medicine, Durham, United States; [3]Center for Host-Microbial Interactions, Duke University School of Medicine, Durham, United States; [4]Department of Physics and Astronomy, University of North Carolina, Chapel Hill, United States

**Abstract** Intermediate filaments (IF) are a major component of the metazoan cytoskeleton and are essential for normal cell morphology, motility, and signal transduction. Dysregulation of IFs causes a wide range of human diseases, including skin disorders, cardiomyopathies, lipodystrophy, and neuropathy. Despite this pathophysiological significance, how cells regulate IF structure, dynamics, and function remains poorly understood. Here, we show that site-specific modification of the prototypical IF protein vimentin with O-linked β-*N*-acetylglucosamine (O-GlcNAc) mediates its homotypic protein-protein interactions and is required in human cells for IF morphology and cell migration. In addition, we show that the intracellular pathogen *Chlamydia trachomatis*, which remodels the host IF cytoskeleton during infection, requires specific vimentin glycosylation sites and O-GlcNAc transferase activity to maintain its replicative niche. Our results provide new insight into the biochemical and cell biological functions of vimentin O-GlcNAcylation, and may have broad implications for our understanding of the regulation of IF proteins in general.
DOI: https://doi.org/10.7554/eLife.31807.001

*For correspondence:
michael.boyce@duke.edu

Competing interests: The authors declare that no competing interests exist.

## Introduction

Intermediate filaments (IF) are a major component of the metazoan cytoskeleton, distinct from the actin and microtubule systems (*Lowery et al., 2015*; *Herrmann and Aebi, 2016*; *Chernyatina et al., 2015*; *Köster et al., 2015*; *Leduc and Etienne-Manneville, 2015*). Humans express over 70 IF proteins, including both cytoplasmic (e.g., vimentin, keratins, neurofilaments) and nuclear (lamins) members, many with tissue-specific functions (*Szeverenyi et al., 2008*). All IF proteins comprise a central, conserved α-helical rod domain, as well as amino-terminal head and carboxy-terminal tail domains of varying lengths (*Lowery et al., 2015*; *Herrmann and Aebi, 2016*; *Chernyatina et al., 2015*; *Köster et al., 2015*; *Leduc and Etienne-Manneville, 2015*). IF proteins homo- or heterodimerize through the parallel association of their rod domains into coiled coils, forming an elongated dimer of ~45–48 nm for cytoplasmic IFs and ~50–52 nm for nuclear lamins (*Quinlan et al., 1986*; *Aebi et al., 1986*). These dimers laterally associate in antiparallel fashion to form tetramers, which in turn assemble into ~65 nm unit-length filaments (ULFs) composed of eight tetramers (*Herrmann and Aebi, 2016*; *Chernyatina et al., 2015*; *Herrmann et al., 1996*). Finally, ULFs associate end-to-end to assemble mature IFs, measuring ~10 nm across (*Lowery et al., 2015*; *Herrmann and Aebi, 2016*; *Chernyatina et al., 2015*).

Unlike actin- or microtubule-based structures, IFs are nonpolar and do not serve as tracks for molecular motors. Instead, IFs contribute to the mechanical integrity of the cell through their unique

**eLife digest** Like the body's skeleton, the cytoskeleton gives shape and structure to the inside of a cell. Yet, unlike a skeleton, the cytoskeleton is ever changing. The cytoskeleton consists of many fibers each made from chains of protein molecules. One of these proteins is called vimentin and it forms intermediate filaments in the cytoskeleton. Many different types of cells contain vimentin and a lot of it is found in cancer cells that have spread beyond their original location to other sites in the body.

Cells use chemical modifications to regulate cytoskeleton proteins. For example, through a process called glycosylation, cells can reversibly attach a sugar modification called O-GlcNAc to vimentin. O-GlcNAc can be attached to several different parts of vimentin and each location may have a different effect. It is not currently clear how cells control their vimentin filaments or what role O-GlcNAc plays in this process.

Using genetic engineering, Tarbet et al. produced human cells in the laboratory with modified vimentin proteins. These altered proteins lacked some of the sites for O-GlcNAc attachment. The goal was to see whether the loss of O-GlcNAc at a specific location would affect fiber formation and cell behavior. The results showed one site where vimentin needs O-GlcNAc to form fibers. Without O-GlcNAc at this site, cells could not migrate towards chemical signals. In addition, in normal human cells, *Chlamydia* bacteria hijack vimentin and rearrange the filaments to form a cage around themselves for protection. However, the cells lacking O-GlcNAc on vimentin were resistant to infection by *Chlamydia* bacteria.

These findings highlight the importance of O-GlcNAc on vimentin in healthy cells and during infection. Vimentin's contribution to cell migration may also help to explain its role in the spread of cancer. The importance of O-GlcNAc suggests it could be a new target for therapies. Yet, it also highlights the need for caution due to the delicate balance between the activity of vimentin in healthy and diseased cells. In addition, human cells produce about 70 other vimentin-like proteins and further work will examine if they are also affected by O-GlcNAc.

DOI: https://doi.org/10.7554/eLife.31807.002

viscoelastic properties (*Lowery et al., 2015*; *Herrmann and Aebi, 2016*; *Chernyatina et al., 2015*; *Köster et al., 2015*; *Leduc and Etienne-Manneville, 2015*). In general, the IF network is flexible under low strain but stiffens and resists breakage under an applied force (*Janmey et al., 1991*; *Fudge et al., 2003*; *Guzmán et al., 2006*; *Kreplak et al., 2005*). Remarkably, individual IFs can be stretched up to 3.6-fold before rupture, demonstrating their elastic nature, as compared to actin cables or microtubules (*Kreplak et al., 2005*). The IF network is also highly dynamic in vivo, with IF subunits (likely tetramers) exchanging rapidly at many points along mature filaments (*Mendez et al., 2010*; *Goldman et al., 2012*; *Miller et al., 1991*; *Vikstrom et al., 1989*; *Ho et al., 1998*; *Martys et al., 1999*; *Vikstrom et al., 1992*; *Nöding et al., 2014*). Similarly, the IF cytoskeleton quickly reorganizes in response to numerous physiological cues, including cell cycle progression, migration, spreading, and growth factor stimulation (*Lowery et al., 2015*; *Herrmann and Aebi, 2016*; *Chernyatina et al., 2015*; *Köster et al., 2015*; *Leduc and Etienne-Manneville, 2015*; *Yoon et al., 1998*; *Helfand et al., 2003*).

IFs participate in many cellular processes, including maintenance of cell shape, organelle anchoring, cell motility, and signal transduction (*Helfand et al., 2011*; *Ben-Ze'ev, 1984*). For example, vimentin, among the most widely studied IF proteins, is required for mesenchymal cell adhesion, migration, chemotaxis, and wound healing in both cell culture and animal models (*Ivaska et al., 2007*; *Yamaguchi et al., 2005*; *Eckes et al., 2000*; *Rogel et al., 2011*; *Menko et al., 2014*). Vimentin IFs also contribute to the mechanical properties of the cytoplasm, and stabilize and localize mitochondria (*Nekrasova et al., 2011*; *Buehler, 2013*; *Guo et al., 2013*; *Eckes et al., 1998*). Importantly, genetic lesions that dysregulate the IF cytoskeleton cause a wide range of human diseases, including skin and nail disorders (keratins), cardiomyopathies (desmin), lipodystrophy, muscular dystrophy and progeria (lamins), giant axonal neuropathy and Charcot-Marie-Tooth disease (neurofilaments), and cataracts (vimentin) (*Omary, 2009*; *Omary et al., 2004*). In addition, the ectopic expression of wild type (WT) vimentin is a hallmark of the epithelial-to-mesenchymal

transition, and is widely observed in human metastatic cancers (*Satelli and Li, 2011*; *Nieto, 2011*; *De Craene and Berx, 2013*). IFs are likely functionally important in this context, because vimentin expression levels correlate with the invasive phenotypes of breast, prostate, and other epithelial cancers (*Satelli and Li, 2011*; *Wei et al., 2008*; *Zhu et al., 2011*; *Vuoriluoto et al., 2011*). Finally, the IF cytoskeleton is also intimately involved in host-microbe interactions (*Geisler and Leube, 2016*; *Mak and Brüggemann, 2016*). For example, changes in vimentin IFs are implicated in the adhesion, invasion, and replication of a wide range of bacteria, including such important pathogens as *Chlamydia trachomatis* (*Kumar and Valdivia, 2008*; *Jorgensen et al., 2011*; *Snavely et al., 2014*; *Bednar et al., 2011*), *Mycobacterium tuberculosis* (*Garg et al., 2006*; *Mahesh et al., 2016*), *Streptococcus pyogenes* (*Bryant et al., 2006*; *Icenogle et al., 2012*; *Lin et al., 2015*) and *Salmonella enterica* (*Murli et al., 2001*; *Guignot and Servin, 2008*).

Despite this broad pathophysiological significance, the regulation of IF cytoskeleton morphology, dynamics, and signaling functions remains incompletely understood. Several recent lines of evidence indicate that post-translational modifications (PTMs) are an important mode of IF regulation, and indeed all IFs are subject to extensive PTMs, including phosphorylation, ubiquitination, sumoylation, acetylation, farnesylation, and glycosylation (*Snider and Omary, 2014*). However, in most cases, the functional impact of these PTMs on IF structure and function is poorly characterized.

To better understand the dynamic regulation of the IF cytoskeleton, we focused on O-linked β-*N*-acetylglucosamine (O-GlcNAc), an intracellular form of protein glycosylation that reversibly decorates serine and threonine residues on thousands of nuclear, cytoplasmic, and mitochondrial proteins. In mammals, O-GlcNAc is added by O-GlcNAc transferase (OGT) and removed by O-GlcNAcase (OGA) (*Figure 1A*) (*Hanover et al., 2010*; *Hart et al., 2011*; *Hart, 2014*). O-GlcNAc cycling controls many processes, including nutrient sensing, cell cycle progression, and apoptosis (*Hanover et al., 2010*; *Hart et al., 2011*), and is essential, as genetic ablation of OGT or OGA is lethal in mice (*Shafi et al., 2000*; *Keembiyehetty et al., 2015*; *Yang et al., 2012*). In addition, aberrant O-GlcNAc cycling is implicated in numerous human diseases, including cancer (*Hart et al., 2011*; *Ma and Vosseller, 2013*; *Yi et al., 2012*; *Singh et al., 2015*), diabetes (*Vaidyanathan and Wells, 2014*; *Hardivillé and Hart, 2014*; *Ma and Hart, 2013*), cardiac dysfunction (*Erickson et al., 2013*; *Erickson, 2014*; *Dassanayaka and Jones, 2014*; *Darley-Usmar et al., 2012*), and neurodegeneration (*Yuzwa et al., 2012*; *Vaidyanathan et al., 2014*; *Yuzwa and Vocadlo, 2014*; *Zhu et al., 2014*).

Interestingly, numerous IF proteins are O-GlcNAcylated (*King and Hounsell, 1989*; *Chou et al., 1992*). For example, keratin-18 glycosylation is required for the recruitment and activation of the pro-survival kinase Akt, and mice expressing an unglycosylatable keratin-18 mutant are sensitized to chemical injury of the liver and pancreas (*Ku et al., 2010*; *Ku et al., 1996*). Several neurofilament proteins are also glycosylated, especially in axons, and neurofilament-M O-GlcNAcylation is reduced in both human Alzheimer's disease patients and a rat model of amyotrophic lateral sclerosis, suggesting a potential role for dysregulated IF glycosylation in neurodegeneration (*Dong et al., 1996*; *Dong et al., 1993*; *Lüdemann et al., 2005*; *Deng et al., 2008*; *Cheung and Hart, 2008*). Finally, vimentin is O-GlcNAcylated on several sites, primarily in its head domain (*Slawson et al., 2008*; *Wang et al., 2007*). Changes in vimentin glycosylation have been observed in models of differentiating adipocytes (*Ishihara et al., 2010*) and neurons (*Farach and Galileo, 2008*), and a recent study observed a correlation between vimentin O-GlcNAcylation and the invasive potential of cholangiocarcinoma (*Phoomak et al., 2017*), implicating glycosylation changes in both the physiological and pathological functions of vimentin. These studies suggest that O-GlcNAcylation is a prominent mode of IF regulation in homeostasis and disease alike. Nevertheless, the mechanistic and functional impacts of O-GlcNAcylation on IF proteins remain largely unexplored.

Here, we report that vimentin O-GlcNAcylation is required for the structure and function of IFs in human cells. We demonstrate that site-specific glycosylation of vimentin mediates its self-association, and is required in human cells for both IF morphology and its facilitating role in cell migration. In addition, we provide evidence that vimentin glycosylation is co-opted by an intracellular bacterial pathogen to produce a replicative niche, revealing a new connection between IF O-GlcNAcylation and microbial pathogenesis. Our results provide new insight into the physiological role of vimentin glycosylation in particular, and may have broad implications for our understanding of the dynamic regulation of the IF cytoskeleton in general.

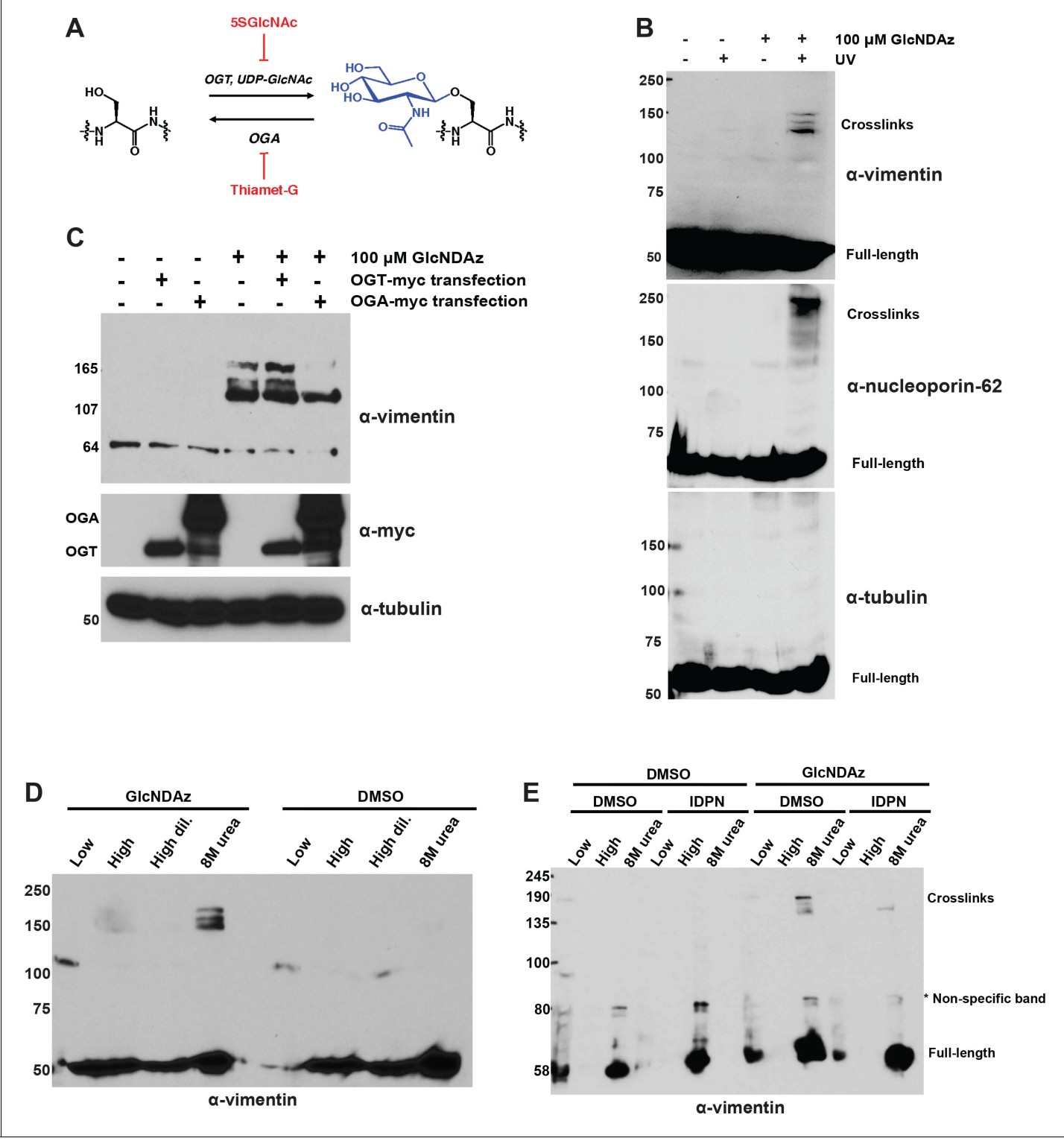

**Figure 1.** Vimentin engages in O-GlcNAc-mediated protein-protein interactions within assembled IFs. (**A**) O-GlcNAc transferase (OGT) uses the nucleotide-sugar donor UDP-GlcNAc to add O-GlcNAc to protein substrates. O-GlcNAcase (OGA) removes O-GlcNAc moieties. 5SGlcNAc and Thiamet-G are specific small molecule inhibitors of OGT and OGA, respectively. (**B**) 293T cells were treated with DMSO vehicle or GlcNDAz (48 hr) and UV light (or not) as indicated, and lysates were prepared in denaturing buffer and analyzed by IB. O-GlcNAc-mediated protein-protein interactions manifest as high molecular weight GlcNDAz-crosslinked complexes (labeled). Heavily glycosylated nucleoporin-62 is a positive control, whereas unglycosylated tubulin is a negative control. Vimentin IB was performed with the D21H3 antibody. (**C**) 293T cells were transfected with OGT-myc or

*Figure 1 continued on next page*

*Figure 1 continued*

OGA-myc constructs, as indicated, and subjected to GlcNDAz crosslinking as above. Crosslinked and uncrosslinked endogenous vimentin species were detected in lysates made in denaturing buffer by IB (D21H3 antibody). (D) 293T cells were subjected to GlcNDAz crosslinking as above. Then, soluble (disassembled) and insoluble (assembled) vimentin populations were separated by differential extraction, as described (*Herrmann et al., 2004*). Low, low ionic strength buffer. High, high dil., high ionic strength buffer (loaded both as-is and diluted, as recommended (*Herrmann et al., 2004*)). 8M urea extracts fully assembled IFs. Crosslinked and uncrosslinked vimentin species were detected by IB (D21H3 antibody). (E) 293T cells were treated with GlcNDAz for 48 hr, treated with 1% IDPN or DMSO vehicle for 30 min, and exposed to UV. Then, cells were subjected to differential extraction, as above, and analyzed by IB (D21H3 antibody). Note that the uncrosslinked vimentin band appears in the 8M urea fraction of IDPN-treated cells because IDPN treatment collapses vimentin IFs into insoluble aggregates (see *Figure 1—figure supplement 2G*) (*Durham, 1986*).

DOI: https://doi.org/10.7554/eLife.31807.003

The following source data and figure supplements are available for figure 1:

**Source data 1.** Proteomic analysis of vimentin GlcNDAz crosslinks.
DOI: https://doi.org/10.7554/eLife.31807.007
**Figure supplement 1.** Analysis of vimentin GlcNDAz crosslinks.
DOI: https://doi.org/10.7554/eLife.31807.004
**Figure supplement 2.** Analysis of vimentin GlcNDAz crosslinks, continued.
DOI: https://doi.org/10.7554/eLife.31807.005
**Figure supplement 3.** Vimentin GlcNDAz crosslinks are not affected by phosphatase treatment.
DOI: https://doi.org/10.7554/eLife.31807.006

## Results

Vimentin is O-GlcNAcylated at multiple sites in its head domain (*Slawson et al., 2008*; *Wang et al., 2007*), but the functional consequences of this modification are poorly understood. Like other intracellular PTMs, O-GlcNAc can exert a wide range of biochemical effects on its substrates, including conformational change, relocalization or destruction (*Hanover et al., 2010*; *Hart et al., 2011*; *Shafi et al., 2000*; *Keembiyehetty et al., 2015*; *Mondoux et al., 2011*; *Bond and Hanover, 2013*). Because IF proteins self-associate into homo-oligomeric complexes, we hypothesized that vimentin O-GlcNAcylation might influence its protein-protein interactions. Indeed, O-GlcNAc regulates protein-protein interactions in other contexts, including chromatin remodeling, nutrient sensing, and the interaction between keratins and Akt (*Ku et al., 2010*; *Fujiki et al., 2011*; *Tarbet et al., 2018*; *Dentin et al., 2008*). However, physiological O-GlcNAc-mediated interactions are often low-affinity, sub-stoichiometric, and transient, presenting a technical barrier to studying them (*Hanover et al., 2010*; *Hart et al., 2011*; *Shafi et al., 2000*; *Keembiyehetty et al., 2015*; *Mondoux et al., 2011*; *Bond and Hanover, 2013*; *Tarbet et al., 2018*).

To address these challenges, we used a chemical biology method to capture and characterize O-GlcNAc-mediated protein-protein interactions in living cells (*Yu et al., 2012*). In this strategy, cells are metabolically labeled with a GlcNAc analog bearing a diazirine photocrosslinking moiety, termed 'GlcNDAz' (*Yu et al., 2012*). GlcNDAz is accepted by the GlcNAc salvage pathway, converted to the nucleotide-sugar UDP-GlcNDAz, and used by OGT to decorate its native substrates (*Yu et al., 2012*). Brief treatment of GlcNDAz-labeled live cells with UV light triggers the covalent crosslinking of O-GlcNDAz moieties to any binding partner proteins within ~2–4 Å of the sugar (*Yu et al., 2012*). Because of this short radius, GlcNDAz crosslinking occurs exclusively at sites where the glycan contributes to the interaction interface, without crosslinking to distant or nonspecific proteins (*Yu et al., 2012*). Therefore, GlcNDAz is a powerful tool for identifying direct, glycosylation-mediated interactions between endogenous proteins in live cells (*Yu et al., 2012*).

To determine whether vimentin engages in O-GlcNAc-mediated protein-protein interactions, we treated human cells with GlcNDAz and UV, and analyzed lysates by immunoblot (IB). Endogenous vimentin crosslinked into distinct, high molecular weight bands that were diazirine- and UV-dependent (*Figure 1B*), indicating that O-GlcNAc mediates protein-protein interactions between vimentin and one or more binding partners. Overexpression of OGT caused an increase in vimentin crosslinks (*Figure 1C*), whereas overexpression of OGA (*Figure 1C*) or treatment with a small molecule inhibitor of OGT, abbreviated 5SGlcNAc (*Gloster et al., 2011*) (*Figure 1—figure supplement 1A*), reduced crosslinking, confirming that this assay reports on authentic O-GlcNAc-mediated interactions, and not a nonspecific action of the GlcNDAz probe.

We next sought to identify the O-GlcNAc-mediated binding partner(s) of vimentin detected in our crosslinking assay. We created a myc-6xHis-tagged human vimentin construct and confirmed that it crosslinked via GlcNDAz similarly to endogenous vimentin (*Figure 1—figure supplement 1B*). Then, we purified preparative amounts of vimentin crosslinks from transfected cells via tandem immunoprecipitation (IP) and immobilized metal affinity chromatography, and analyzed the crosslinks by mass spectrometry (MS)-based proteomics (*Figure 1—figure supplement 1C*, *Figure 1—figure supplement 2D,E* and *Figure 1—source data 1*). We obtained 80% tryptic peptide coverage of vimentin in these samples, without significant enrichment of other proteins, beyond common contaminants (*Figure 1—figure supplement 2* and *Figure 1—source data 1*). This result suggested that the vimentin GlcNDAz crosslinks represent homotypic, O-GlcNAc-mediated vimentin-vimentin interactions.

In vivo, vimentin exists in a range of assembly states, from soluble tetramers and ULFs, to relatively insoluble mature IFs (*Lowery et al., 2015*; *Herrmann and Aebi, 2016*; *Chernyatina et al., 2015*; *Köster et al., 2015*; *Leduc and Etienne-Manneville, 2015*). To determine the assembly state of the crosslinked vimentin species, we performed a well-established differential extraction procedure (*Ridge et al., 2016*) on GlcNDAz-crosslinked cells. The GlcNDAz crosslinks of endogenous vimentin extracted into a denaturing buffer but not low or high ionic strength non-denaturing buffers, indicating that the crosslinks occur within the highly assembled filament population (*Figure 1D*) (*Ridge et al., 2016*). Importantly, extracted crosslinks remained soluble when exchanged from denaturing into non-denaturing buffers, demonstrating that GlcNDAz crosslinking per se does not reduce vimentin solubility (*Figure 1—figure supplement 2F*). To further confirm that crosslinks occur within assembled IFs, we treated cells with β,β'-iminodipropionitrile (IDPN), which blocks vimentin assembly beyond the ULF state (*Durham, 1986*). Consistent with previous reports, IDPN treatment abrogated vimentin IFs and caused the accumulation of vimentin aggregates (*Figure 1—figure supplement 2G*) (*Durham, 1986*). In the crosslinking assay, IDPN also suppressed the formation of GlcNDAz-dependent adducts, indicating that the crosslinks occur in assembly states beyond ULFs (*Figure 1E*). Using quantitative IBs, we detected ~10% of endogenous vimentin crosslinked into GlcNDAz-dependent adducts (*Figure 1—figure supplement 2H*), though this measurement likely significantly underestimates the fraction of vimentin that engages in O-GlcNAc-mediated interactions, since several steps in the GlcNDAz crosslinking protocol are less than 100% efficient (*Yu et al., 2012*; *Rodriguez et al., 2015*). Based on these results, we concluded that O-GlcNAc-mediated vimentin-vimentin interactions are prevalent in cells, and occur primarily within assembled IFs, but not in soluble pools of lower-order assembly states.

The Hart lab previously mapped several glycosylation sites on vimentin's flexible head domain (*Slawson et al., 2008*; *Wang et al., 2007*). We used these results to identify vimentin O-GlcNAc sites that contribute to its homotypic, glycosylation-mediated interactions. We mutated each reported O-GlcNAcylation site to alanine and screened these constructs in the GlcNDAz crosslinking assay (*Figure 2*). Mutation of several individual residues, including T33, S34, and S39, reduced vimentin crosslinking, while mutation of S49 abolished all detectable crosslinks. Because of the dramatic reduction of crosslinking in the S49A mutant, we measured the fraction of total vimentin glycosylation occurring at S49. We transfected cells with vector, WT or S49A vimentin and then labeled them with GalNAz, an azide-bearing unnatural monosaccharide that we have described previously as a metabolic reporter of O-GlcNAcylation (*Boyce et al., 2011*; *Palaniappan et al., 2013*; *Chen et al., 2017*). S49A mutant vimentin exhibited ~80% the GalNAz signal of WT (*Figure 2—figure supplement 1*), suggesting that approximately one-fifth of vimentin glycosylation occurs on S49 under these conditions. Together, these results suggest that robust and site-specific O-GlcNAcylation in the vimentin head domain mediates homotypic protein-protein interactions within IFs.

We next tested whether O-GlcNAc-mediated interactions influence vimentin IF structure or function in live human cells. We used CRISPR/Cas9 methods to delete endogenous vimentin from two different human cell lines, selected single cell-derived clones, and confirmed the absence of vimentin mRNA and protein (*Figure 3—figure supplement 1A,B*). Then, we stably reconstituted individual vimentin$^{-/-}$ clones with WT or unglycosylatable point-mutant versions of a well-characterized vimentin-mEmerald fusion protein (*Mendez et al., 2010*; *Yoon et al., 1998*; *Helfand et al., 2011*; *Hookway et al., 2015*) to permit live-cell visualization of IFs. We verified that vimentin-mEmerald was expressed at uniform levels comparable to endogenous vimentin (*Figure 3—figure supplement 1C,D*), and that the mEmerald signal precisely coincided with anti-vimentin immunofluorescence in

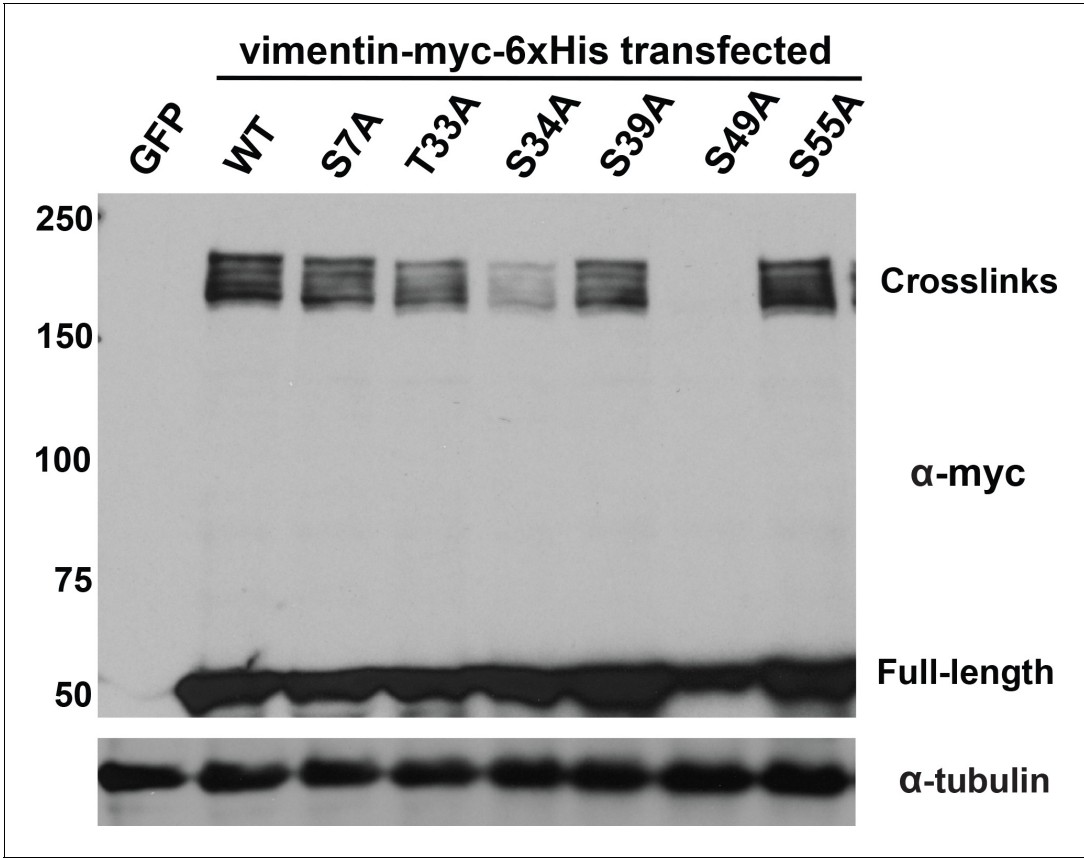

**Figure 2.** Specific glycosylation sites in the vimentin head domain are required for homotypic, O-GlcNAc-mediated interactions. 293T cells were transfected with GFP only (control) or WT or mutant vimentin-myc-6xHis constructs as indicated for 24 hr, subjected to GlcNDAz crosslinking, and analyzed by IB. Tubulin is a loading control.

DOI: https://doi.org/10.7554/eLife.31807.008

The following figure supplements are available for figure 2:

**Figure supplement 1.** Quantitative examination of S49 glycosylation.
DOI: https://doi.org/10.7554/eLife.31807.009

**Figure supplement 2.** Mutations in putative extracellular GlcNAc-binding sites of vimentin do not reduce intracellular GlcNDAz crosslinking.
DOI: https://doi.org/10.7554/eLife.31807.010

cells expressing both endogenous vimentin and vimentin-mEmerald (*Figure 3—figure supplement 2E*). These results confirmed that the vimentin-mEmerald construct is a faithful proxy for the untagged protein in this system. In our panels of vimentin-mEmerald-reconstituted cells, we included Y117L, a point-mutation in the rod domain that blocks vimentin assembly beyond the ULF stage (*Meier et al., 2009*), as a positive control for IF disruption. We used fluorescence-activated cell sorting, IB and fluorescence microscopy to ensure equal expression of the various WT and mutant vimentin-mEmerald transgenes across reconstituted vimentin$^{-/-}$ cell lines (*Figure 3A*, *Figure 3—figure supplements 1D,3A*).

Cells reconstituted with WT vimentin-mEmerald exhibited canonical IF morphology, whereas the Y117L-expressing cells lacked assembled filaments and instead displayed punctate structures consistent with ULFs (*Figure 3A,B*, *Figure 3—figure supplement 3A,B*). These results indicate that our reconstituted cell systems recapitulate the previously reported characteristics of WT and Y117L vimentin (*Meier et al., 2009*; *Helfand et al., 2011*; *Robert et al., 2014*). We also observed dramatic alterations in vimentin IF organization in several unglycosylatable point-mutants. For example, the S34A mutant, which had an intermediate phenotype in our crosslinking assay (*Figure 2*), displayed a partial defect in IF formation, with both punctate and filamentous fluorescence detected (*Figure 3A*, *Figure 3—figure supplement 3A*). Interestingly, the S49A mutant, which lacked detectable

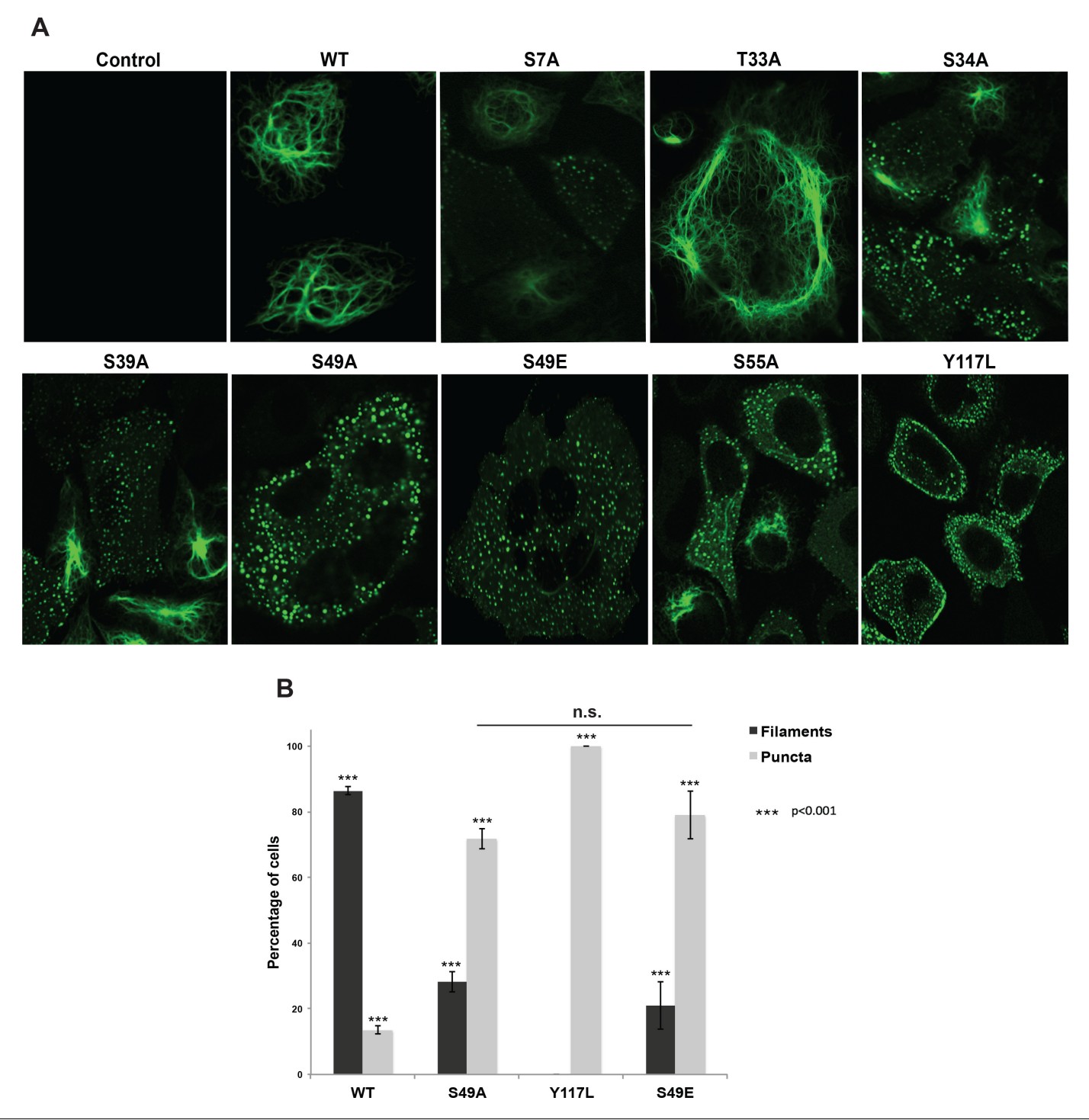

**Figure 3.** Head domain glycosylation sites are required for vimentin IF assembly and morphology in vivo. (**A**) A vimentin$^{-/-}$ HeLa clone was stably transduced with empty vector (control) or expression constructs encoding WT or mutant vimentin-mEmerald and then imaged by laser scanning confocal fluorescence microscopy. (**B**) Vimentin-reconstituted HeLa cells were imaged as in (**A**) and cells were scored for filaments or puncta (≥400 cells per genotype). Differences in both filament and puncta measurements are significant across all genotype comparisons (***$p<0.001$, Student's t-test) except for S49A and S49E, which are indistinguishable from each other.

DOI: https://doi.org/10.7554/eLife.31807.011

The following figure supplements are available for figure 3:

**Figure supplement 1.** Construction and validation of vimentin-reconstituted cells.

*Figure 3 continued on next page*

*Figure 3 continued*

DOI: https://doi.org/10.7554/eLife.31807.012

**Figure supplement 2.** S49A mutation does not affect vimentin stability.

DOI: https://doi.org/10.7554/eLife.31807.013

**Figure supplement 3.** Head domain glycosylation sites are required for vimentin IF assembly and morphology in vivo.

DOI: https://doi.org/10.7554/eLife.31807.014

O-GlcNAc-mediated interactions in the GlcNDAz assay (*Figure 2*), exhibited a significant reduction of assembled IFs in live cells, displaying a higher proportion of punctate vimentin fluorescence as compared to WT (*Figure 3A,B*, *Figure 3—figure supplement 3A,B*). Importantly, the S49A mutation had no impact on vimentin stability (*Figure 3—figure supplement 2*), further ruling out an effect of vimentin expression level in this system.

These results suggested that O-GlcNAc on specific residues of vimentin, particularly S49, may be required for its homotypic assembly into mature IFs in live cells. However, S/T→A mutations eliminate all O-linked PTMs at the mutated site, including both O-GlcNAcylation and phosphorylation. Therefore, phenotypes arising from S/T→A mutations may be due to loss of phosphorylation, glycosylation, or both. Vimentin S49 is a known glycosylation site (*Slawson et al., 2008*; *Wang et al., 2007*) but not a reported phosphorylation site, suggesting that the phenotypes we observed in the S49A mutant (*Figures 2* and *3*, *Figure 3—figure supplement 3*) are due to the loss of O-GlcNAc, and not the loss of phosphorylation. To further test this hypothesis, we reconstituted vimentin$^{-/-}$ cells with a phosphomimetic S49E mutant. Vimentin-S49E-expressing cells displayed a loss of filament morphology and a proportion of puncta indistinguishable from the S49A mutant (*Figure 3* and *Figure 3—figure supplement 3*). These results suggest that the abnormal vimentin IF structures observed with the S49A mutant in live cells are due to the loss of glycosylation, not loss of phosphorylation, at S49.

We next investigated whether vimentin O-GlcNAcylation is required for known functions of IFs. The IF cytoskeleton facilitates cell migration, and vimentin$^{-/-}$ cells and tissues exhibit migration defects (*Ivaska et al., 2007*; *Yamaguchi et al., 2005*; *Eckes et al., 2000*; *Rogel et al., 2011*; *Menko et al., 2014*). We used a well-characterized Transwell assay (*Justus et al., 2014*) to measure cell migration by vimentin-reconstituted cell lines across a collagen matrix. Consistent with prior reports in other systems (*Ivaska et al., 2007*; *Yamaguchi et al., 2005*; *Eckes et al., 2000*; *Rogel et al., 2011*; *Menko et al., 2014*), vimentin$^{-/-}$ cells stably transduced with empty vector or the Y117L mutant construct were impaired in serum-stimulated migration, relative to cells expressing WT vimentin (*Figure 4*, *Figure 4—source data 1*). Interestingly, vimentin-S49A-expressing cells also exhibited migration defects compared to WT (*Figure 4*, *Figure 4—source data 1*). This result suggested that O-GlcNAc on vimentin may be required for optimal cell migration. To further confirm this hypothesis, we assayed the serum-induced migration of vimentin-reconstituted cells upon treatment with 5SGlcNAc or Thiamet-G, a specific small molecule inhibitor of OGA (*Yuzwa et al., 2008*) (*Figure 4*, *Figure 4—source data 1*). 5SGlcNAc or Thiamet-G treatment each significantly inhibited the serum-induced migration of cells expressing WT vimentin, but not cells lacking vimentin or expressing the S49A mutant (*Figure 4*, *Figure 4—source data 1*). Taken together, these results indicate that O-GlcNAcylation of vimentin, especially on S49, is required for optimal serum-stimulated cell migration.

Several intracellular pathogens co-opt the IF cytoskeleton to stabilize the large membrane-bound vacuoles in which they replicate, presumably by providing structural scaffolds (*Geisler and Leube, 2016*; *Mak and Brüggemann, 2016*). For example, we previously showed that the obligate intracellular pathogen *Chlamydia trachomatis* recruits a meshwork of vimentin IFs to its vacuolar 'inclusion' compartment, stabilizing its replicative niche and shielding bacterial components from the host's cytoplasmic innate immune surveillance machinery (*Kumar and Valdivia, 2008*; *Jorgensen et al., 2011*; *Snavely et al., 2014*; *Bednar et al., 2011*). The molecular mechanisms of *Chlamydia*-induced vimentin recruitment and reorganization are incompletely understood. To examine the potential role of vimentin O-GlcNAcylation in these processes, we treated HeLa cells reconstituted with WT vimentin-mEmerald with vehicle control, 5SGlcNAc or Thiamet-G, infected them with *Chlamydia*, and visualized pathogen-containing inclusions and vimentin IFs by fluorescence microscopy (*Figure 5A*, *Figure 5—figure supplement 1*). We found that 5SGlcNAc treatment inhibited the formation of a

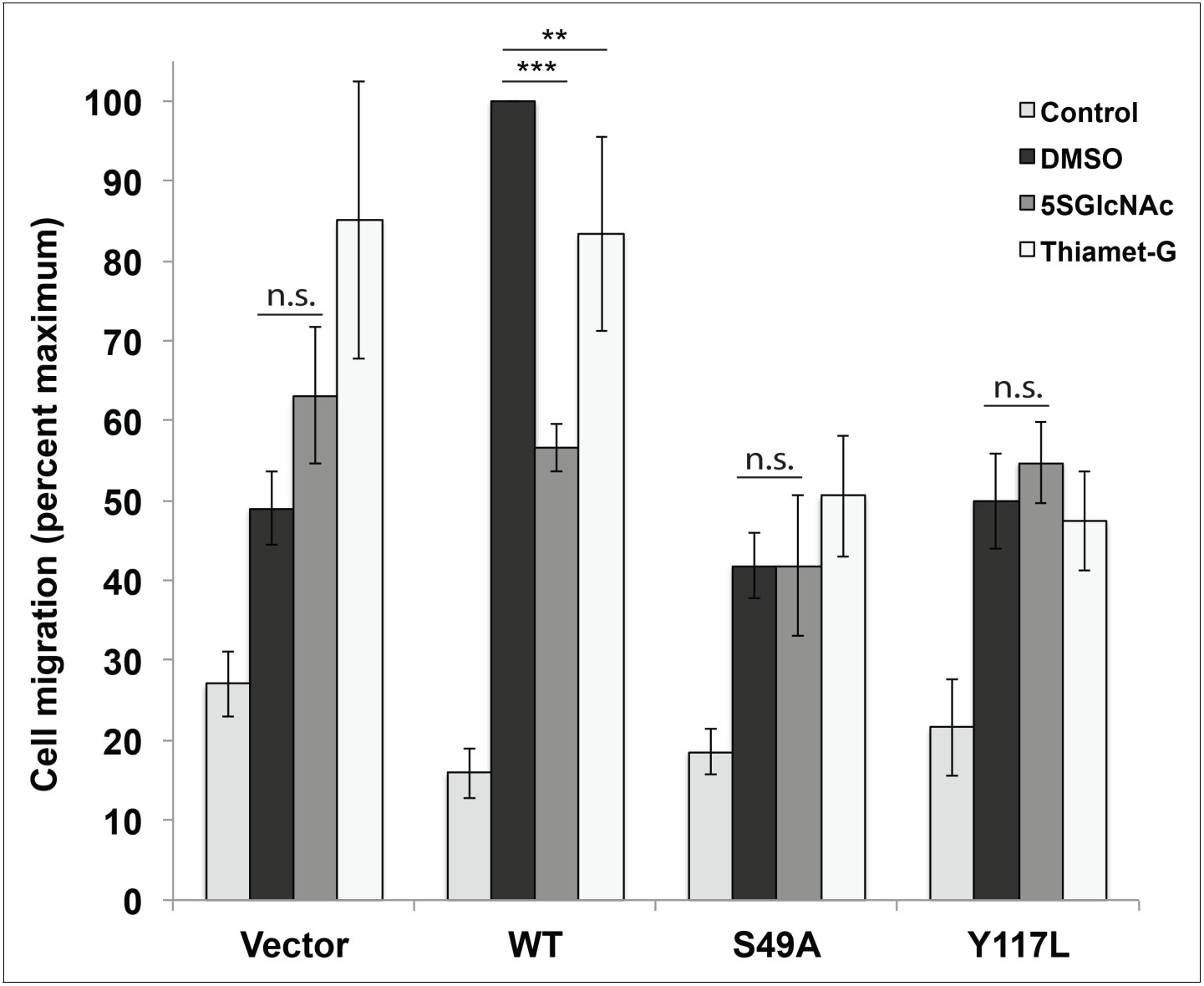

**Figure 4.** Vimentin O-GlcNAcylation is required for cell migration. Vimentin$^{-/-}$ HeLa cells reconstituted with empty vector or WT, S49A or Y117L vimentin-mEmerald expression constructs were serum-starved for 72 hr, treated with either vehicle control (DMSO), 100 µM 5SGlcNAc or 50 µM Thiamet-G for 6 hr, and then assayed by Transwell migration using 10% fetal bovine serum as a chemoattractant (or no serum, 'Control'). Migrated cells were stained with crystal violet and four fields of view were imaged and counted for each of four biological replicates. The WT DMSO serum-stimulated sample was defined as maximum migration and used to normalize all data. Serum-stimulated migration is impaired in cells lacking vimentin or expressing mutant vimentin. 5SGlcNAc and Thiamet-G each inhibit migration in cells expressing WT vimentin, but have no effect on cells lacking vimentin or expressing mutant vimentin. n = 4, ***p<0.001, **p=0.006, n.s. not significant, ANOVA followed by Student's t-test. For simplicity, only selected statistical comparisons are indicated on the graph. Please see *Figure 4—source data 1* for comprehensive statistical comparisons.
DOI: https://doi.org/10.7554/eLife.31807.015

The following source data is available for figure 4:

**Source data 1.** Full statistical analyses of cell migration data.
DOI: https://doi.org/10.7554/eLife.31807.016

vimentin meshwork around the inclusion, consistent with a requirement for vimentin glycosylation in IF remodeling during *Chlamydia* infection (*Figure 5A*). 5SGlcNAc treatment reduced the average size of the inclusions in infected cells and increased the number of extra-inclusion bacteria (*Figure 5B*), demonstrating that OGT activity is required for inclusion expansion and integrity.

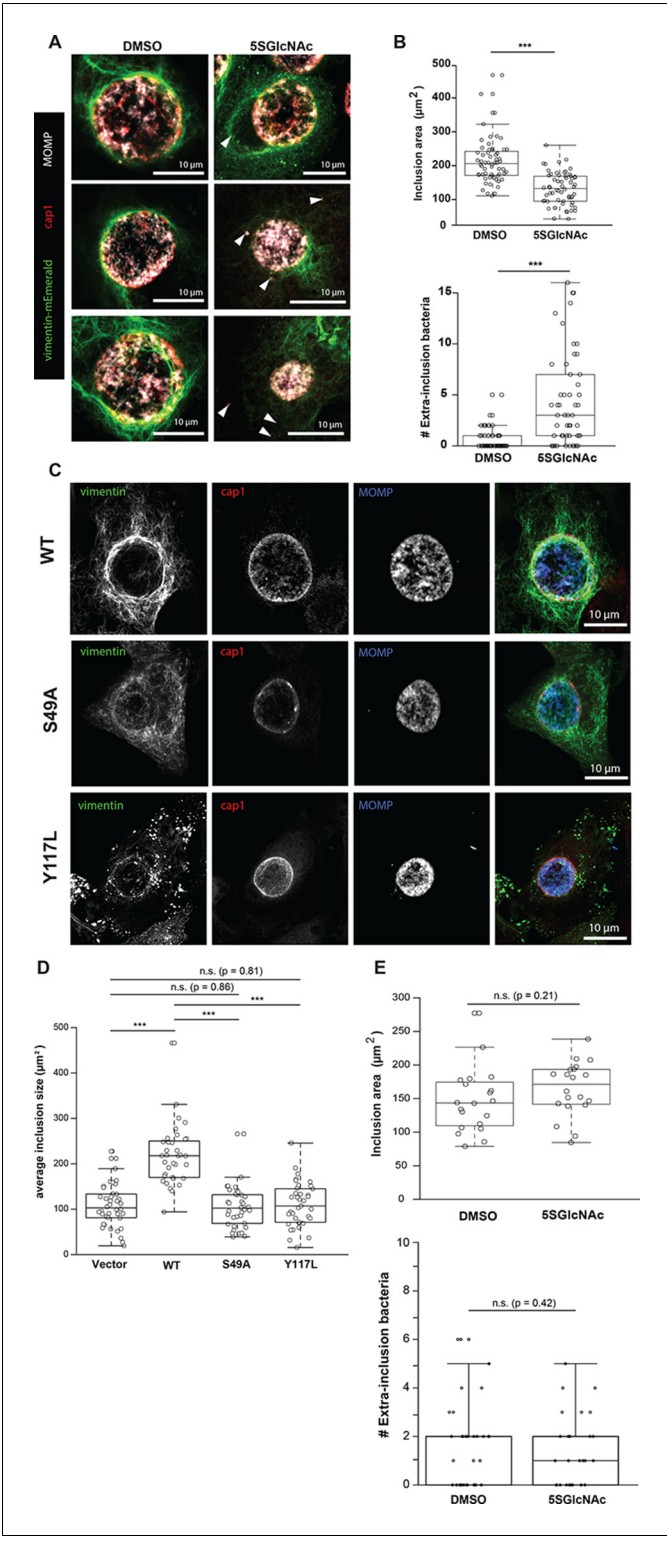

**Figure 5.** *Chlamydia trachomatis* requires OGT activity and vimentin glycosylation sites to maintain inclusion integrity during infection. (**A**) HeLa cells reconstituted with WT vimentin-mEmerald were infected with *Chlamydia* for ten hours and treated with DMSO vehicle or 50 µM 5SGlcNAc for an additional 20 hr. Cells were fixed and immunostained for MOMP (to mark individual bacteria) and cap1 (to mark the inclusion membrane), along with vimentin-mEmerald imaging. Representative images are shown. Arrows indicate extra-inclusion bacteria. (**B**) Quantification of inclusion area and number of extra-inclusion bacteria from images in (**A**). n = 60, ***p<0.001,

*Figure 5 continued on next page*

*Figure 5 continued*

Welch's t-test. (**C**) Reconstituted HeLa cell lines were infected with *Chlamydia* for 30 hr and then fixed, stained, and imaged as in (**A**). Representative images are shown. (**D**) Quantification of inclusion area from images in (**C**). n ≥ 54, ***p<0.001, Welch's t-test. n.s., not significant. (**E**) HeLa cells reconstituted with empty vector were infected with *Chlamydia*, treated with DMSO vehicle or 50 µM 5SGlcNAc, and fixed and stained as in (**A**). Inclusion size (top; n = 20, p=0.21, Student's t-test) and extra-inclusional bacteria (bottom; n = 28, p=0.42, Student's t-test) were quantified. n.s., not significant.

DOI: https://doi.org/10.7554/eLife.31807.017

The following figure supplements are available for figure 5:

**Figure supplement 1.** No effect of Thiamet-G treatment on *Chlamydia* inclusion size.
DOI: https://doi.org/10.7554/eLife.31807.018

**Figure supplement 2.** The S49A mutation does not prevent *Chlamydia*-induced vimentin cleavage.
DOI: https://doi.org/10.7554/eLife.31807.019

---

Thiamet-G treatment had little effect in these assays, likely because basal levels of O-GlcNAcylation are already sufficient for optimal *Chlamydia* replication (*Figure 5*, *Figure 5—figure supplement 1*).

We next tested whether O-GlcNAcylation of vimentin itself is required for inclusion integrity. We infected vimentin-reconstituted HeLa cell lines with *Chlamydia* and visualized vimentin IFs and bacteria by fluorescence microscopy. Compared to WT vimentin-expressing cells, cells expressing either the S49A or Y117L mutant vimentin displayed reduced IF recruitment to the inclusions, smaller average inclusion size, and larger numbers of bacteria escaping into the cytoplasm (*Figure 5C,D*). These data suggest that the site-specific glycosylation of vimentin itself is required for IF remodeling during *Chlamydia* infection. To further test this hypothesis, we infected empty vector-reconstituted vimentin$^{-/-}$ cells with *Chlamydia* in the presence or absence of 5SGlcNAc. In contrast to our observations with cells expressing WT vimentin (*Figure 5A,B*), 5SGlcNAc had no impact on *Chlamydia* inclusion size or extra-inclusion bacteria in cells lacking vimentin, indicating that vimentin, but not other host- or pathogen-encoded targets, is required for the effects of 5SGlcNAc in this context (*Figure 5E*). We concluded that both OGT activity and vimentin glycosylation sites are required for IF reorganization during *Chlamydia* infection, and are co-opted by this pathogen to promote inclusion integrity and growth.

## Discussion

The IF cytoskeleton plays critical roles in both physiological and pathological processes, but how cells regulate IF structure and function remains poorly understood. We provide evidence that site-specific glycosylation of the vimentin head domain regulates its homotypic association in human cells, and is required for both IF morphology and cell migration (*Figure 6*). In addition, site-specific O-GlcNAcylation of vimentin is exploited by an intracellular pathogen to promote its own replication, underlining the importance of IF dynamics in disease states. Because many IF proteins are O-GlcNAc-modified (*King and Hounsell, 1989*; *Chou et al., 1992*; *Ku et al., 2010*; *Dong et al., 1996*; *Dong et al., 1993*; *Lüdemann et al., 2005*; *Deng et al., 2008*; *Cheung and Hart, 2008*; *Slawson et al., 2008*; *Wang et al., 2007*; *Srikanth et al., 2010*; *Kakade et al., 2016*; *Tao et al., 2006*), our results may provide new insight into the regulation of both vimentin in particular and the IF cytoskeleton in general.

Early evidence for vimentin glycosylation was reported nearly twenty-five years ago (*Shikhman et al., 1993*) and confirmed with sophisticated MS methods thirteen years later (*Slawson et al., 2008*; *Wang et al., 2007*). However, the biochemical and cellular effects of this modification have remained largely uncharacterized. Through GlcNDAz crosslinking (*Figure 2*) and live-cell imaging (*Figure 3*, *Figure 3—figure supplement 3*), we determined that the vimentin glycosylation sites S34, S39 and especially S49 are required for normal homotypic vimentin-vimentin interactions and for IF morphology in live human cells. These O-GlcNAc-mediated interactions likely occur within assembled IFs and not smaller oligomeric states (*Figure 1D,E*). It is well established that portions of the vimentin head domain are required for filament assembly (*Eriksson et al., 2004*), and that head domain PTMs (especially phosphorylation) govern IF dynamics in vivo (*Helfand et al., 2011*; *Eriksson et al., 2004*; *Chou et al., 1990*; *Eriksson et al., 1992*; *Sihag et al.,*

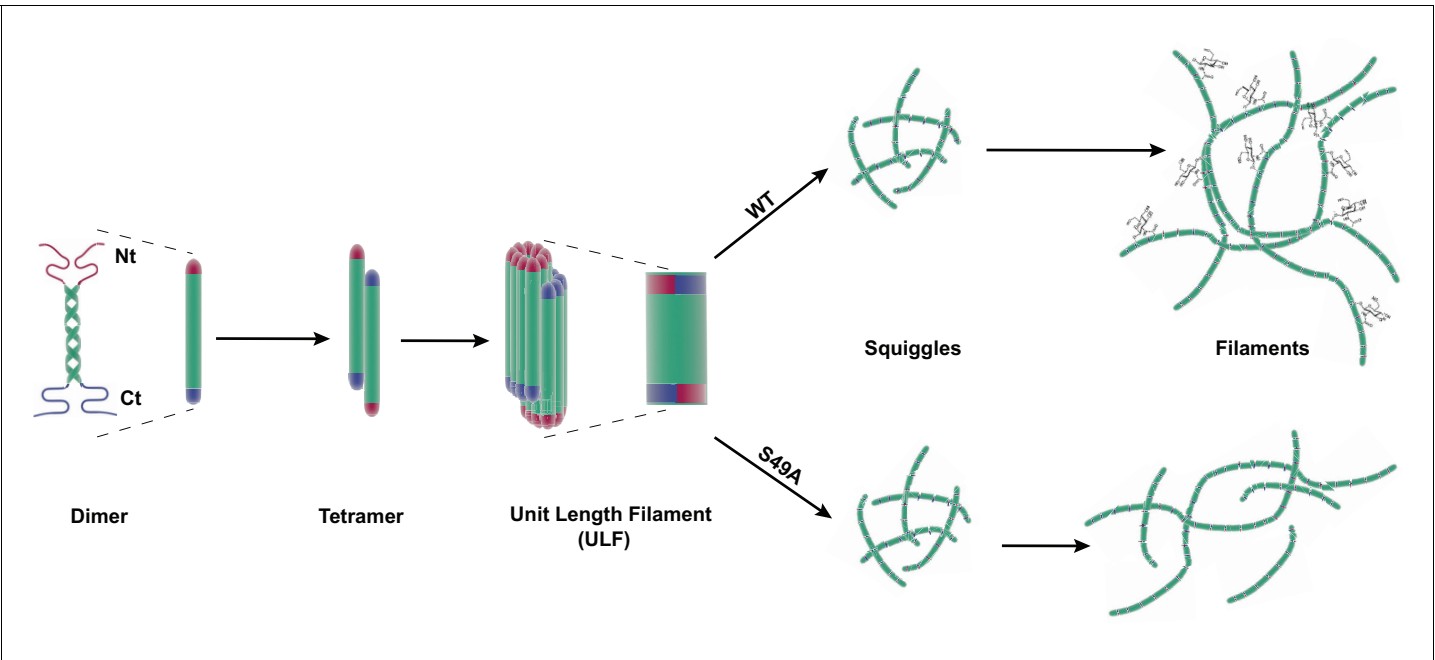

**Figure 6.** Site-specific glycosylation regulates the form and function of vimentin IFs. Our data suggest a model wherein glycosylation of the N-terminal vimentin head domain (red), particularly on S49, promotes homotypic vimentin-vimentin interactions, and assembly and/or maintenance of mature IFs under both homeostatic and *Chlamydia* infection conditions.

DOI: https://doi.org/10.7554/eLife.31807.020

The following figure supplement is available for figure 6:

**Figure supplement 1.** Partial conservation of S49 among vimentin orthologs and human type III IF proteins.

DOI: https://doi.org/10.7554/eLife.31807.021

*2007*; *Goto et al., 2002*; *Chan et al., 2002*). Our results are consistent with this general model of regulated filament assembly/disassembly through head domain PTMs (*Figure 6*). Interestingly, however, internal deletion mutations within the head domain demonstrate that S49 itself is not required for vimentin filament formation in vitro or in vivo (*Shoeman et al., 2002*). Therefore, we propose that the phenotypes observed in the S49A vimentin mutant are caused by a loss of PTM regulation, rather than simply a requirement for serine at that site.

Given the very short crosslinking radius of the GlcNDAz reagent (~2–4 Å) (*Yu et al., 2012*) and our MS proteomics analysis of the crosslinked complexes (*Figure 1—figure supplement 1C* and *Figure 1—figure supplement 2D,E*), the O-GlcNAc-mediated interactions we observe are likely homotypic dimeric or trimeric vimentin adducts. The precise molecular nature of these crosslinks, and why we observe multiple discrete crosslinked complexes, remain uncertain. The high MS proteomic coverage we obtained of vimentin and the lack of other proteins in the crosslinks (*Figure 1—figure supplement 2* and *Figure 1—source data 1*) suggest that the complexes do not contain other binding partner proteins or known protein-based PTMs of vimentin, such as SUMO (*Kaminsky et al., 2009*; *Wang et al., 2010*). Because vimentin is multiply phosphorylated (*Helfand et al., 2011*; *Eriksson et al., 2004*; *Chou et al., 1990*; *Eriksson et al., 1992*; *Sihag et al., 2007*; *Goto et al., 2002*; *Chan et al., 2002*), the crosslinks of distinct molecular weight (*Figures 1B* and *2*) could in principle represent different phospho-forms of vimentin. However, several commercially available anti-phospho-vimentin antibodies failed to recognize the crosslinked complexes by IB (not shown), and phosphatase treatment of crosslinked adducts had no effect on their SDS-PAGE migration (*Figure 1—figure supplement 3*), arguing against differential phosphorylation as an explanation.

Instead, we postulate that the discrete bands in the GlcNDAz-induced vimentin complexes represent different crosslinking geometries of vimentin multimers. Because the O-GlcNAc-mediated interactions we detect occur primarily in assembled IFs (*Figure 1D,E*), an individual glycan on the vimentin head domain (e.g., at residue S49) may contact the head domain of its dimeric partner,

and/or the rod domain of a vimentin molecule in an adjacent dimer, and/or another vimentin molecule within the same ULF. These different crosslinking geometries might produce migration differences in our GlcNDAz/SDS-PAGE assay (*Figure 1B*).

Irrespective of the molecular identities of the individual adducts, the fact that we observe a highly reproducible crosslinking pattern suggests that O-GlcNAc mediates specific contacts among vimentin molecules, and not a large variety of nonspecific interactions, which would produce variable or smeared crosslinking (*Yu et al., 2012*). This evidence of specific contacts is perhaps surprising, given the widely accepted model that the head domain of vimentin is intrinsically disordered (*Guharoy et al., 2013*). It is possible that, in vivo, O-GlcNAcylation mediates site-specific contacts between glycans on one vimentin molecule and discrete glycan recognition sites or domains on adjacent vimentin molecules within assembled filaments. Interestingly, several prior reports identified a GlcNAc-binding property of vimentin on the cell surface (*Ise et al., 2010*; *Ise et al., 2011*; *Kim et al., 2012*; *Komura et al., 2012*). Although the physiological relevance of extracellular vimentin is controversial, these biochemical observations are reminiscent of our own, in that vimentin is reported to bind to GlcNAc residues in both cases. However, two mutations in vimentin that abolish extracellular GlcNAc binding, E382A and E396A (*Komura et al., 2012*), do not reduce intracellular GlcNDAz crosslinking (*Figure 2—figure supplement 2*). Therefore, our observations are distinct from the reported glycan-binding property of cell surface vimentin (*Ise et al., 2010*; *Ise et al., 2011*; *Kim et al., 2012*; *Komura et al., 2012*). The vimentin site(s) that bind O-GlcNAc moieties are not readily discernible from our MS studies, due to the unpredictable masses and fragmentation spectra of crosslinked adducts. However, the identification of these sites will be an important priority for future studies, because this information may elucidate the biophysical regulation of IF assembly and dynamics by O-GlcNAc.

At the cellular level, specific glycosylation sites of vimentin, including S34, S39 and especially S49, are essential for IF morphology and for facilitating serum-stimulated migration (*Figures 3* and *4*, *Figure 3—figure supplement 3*). We propose that O-GlcNAcylation at one or more of these sites influences the assembly and/or disassembly of vimentin filaments, as has been described for other PTMs. Indeed, phosphorylation of several residues in the vimentin head domain promotes IF disassembly in response to multiple stimuli (*Helfand et al., 2011*; *Eriksson et al., 2004*; *Chou et al., 1990*; *Eriksson et al., 1992*; *Sihag et al., 2007*; *Goto et al., 2002*; *Chan et al., 2002*). O-GlcNAcylation and phosphorylation often compete for nearby or identical residues on specific protein substrates, giving rise to a complex functional interplay between these PTMs (*Hart et al., 2011*; *Butkinaree et al., 2010*). Moreover, specific O-GlcNAc sites required for in vivo IF assembly in our work, such as S34 and S39 (*Figure 3*, *Figure 3—figure supplement 3*), can also be phosphorylated (*Helfand et al., 2011*; *Eriksson et al., 2004*; *Chou et al., 1990*; *Eriksson et al., 1992*; *Sihag et al., 2007*; *Goto et al., 2002*; *Chan et al., 2002*). Therefore, reciprocal PTMs at certain sites may have antagonistic effects, with glycosylation of S34 or S39 promoting IF assembly, while phosphorylation at those sites induces disassembly.

In contrast, vimentin S49 is required for normal IF assembly in human cells (*Figures 2* and *3*, *Figure 3—figure supplement 3*) and is a known glycosylation site (*Slawson et al., 2008*; *Wang et al., 2007*) but is not a reported phosphorylation site. These observations suggest a functional role for O-GlcNAc, but not O-phosphate, at this residue. In support of this hypothesis, a phosphomimetic S49E mutant phenocopies an S49A mutant in IF morphology (*Figure 3*), and an OGT inhibitor impacts on cell migration and *Chlamydia* inclusion size only in the presence of WT vimentin, but not in cells expressing S49A mutant vimentin or lacking vimentin (*Figures 4* and *5E*). Taken together, these results strongly indicate that (de)glycosylation of S49 is required for WT levels of cell migration and pathogen-induced IF remodeling. Although we cannot strictly rule out the possibility that S49 is phosphorylated under as-yet untested conditions, we suggest that S49 O-GlcNAcylation is required for homotypic vimentin-vimentin interactions (*Figure 2*), filament assembly (*Figure 3*), and downstream behaviors of the vimentin IF cytoskeleton in normal and pathological contexts (*Figures 4* and *5*).

Finally, our results indicate that O-GlcNAc-mediated vimentin IF assembly is required for *Chlamydia* inclusion expansion and integrity, because we observed smaller inclusions and more cytoplasmic bacteria in *Chlamydia*-infected cells expressing Y117L or S49A mutants of vimentin (*Figure 5C,D*). *Chlamydia* remodels the IF cytoskeleton to stabilize the inclusion during its replication phase, and deploys the CPAF protease to dismantle the IF cytoskeleton and promote bacterial

release at late stages of infection (*Kumar and Valdivia, 2008*; *Jorgensen et al., 2011*; *Snavely et al., 2014*; *Bednar et al., 2011*). Our current data extend these results by demonstrating that vimentin filament remodeling by *Chlamydia* depends on both OGT activity (*Figure 5A,B*) and on the site-specific glycosylation of vimentin itself (*Figure 5C–E*). We detected WT levels of CPAF-mediated vimentin cleavage with the S49A and Y117L mutants (*Figure 5—figure supplement 2*), indicating that the phenotypes we observe are not due to differential vimentin cleavage during infection. The molecular mechanisms by which *Chlamydia* co-opts OGT activity and vimentin glycosylation sites (e.g., S49) to reshape the IF cytoskeleton remain to be determined, and will be an interesting focus of future studies.

In conclusion, we have shown that site-specific glycosylation of vimentin mediates its homotypic protein-protein interactions, and is required in human cells for IF morphology and cell migration. In addition, *Chlamydia* relies on OGT activity and particular vimentin glycosylation sites to remodel the IF cytoskeleton and promote inclusion expansion and integrity. Our results provide new insight into the biochemical and cell biological functions of vimentin O-GlcNAcylation. In addition, our work may have broad implications for other IF proteins. Numerous IF proteins are dynamically glycosylated *in vivo* (*King and Hounsell, 1989*; *Chou et al., 1992*; *Ku et al., 2010*; *Dong et al., 1996*; *Dong et al., 1993*; *Lüdemann et al., 2005*; *Deng et al., 2008*; *Cheung and Hart, 2008*; *Slawson et al., 2008*; *Wang et al., 2007*; *Srikanth et al., 2010*; *Kakade et al., 2016*; *Tao et al., 2006*), and the S49 residue of vimentin is conserved both among vertebrate vimentin orthologs and in human desmin, another type III IF protein (*Figure 6—figure supplement 1*). Therefore, site-specific O-GlcNAcylation may be a general mode of regulating IF dynamics and function. In the future, pharmacological modulation of O-GlcNAcylation may provide a way to manipulate filament form and function for therapeutic benefit in diseases of dysregulated IFs, ranging from neurodegeneration and cardiomyopathy to cancer.

## Materials and methods

### Key resources table

| Reagent type (species) or resource | Designation | Source or reference | Identifiers | Additional information |
|---|---|---|---|---|
| Cell line (HeLa) | vimentin -/- | this paper | | |
| Cell line (293T/17) | 293T vimentin -/- | this paper | | |
| Cell line (HeLa) | HeLa | | | |
| Cell line (293T/17) | 293T/UAP1 | PMCID: PMC3323966 | | Stably expressing AGX1(F383G) |
| Antibody | Anti-O-GlcNAc RL2 | Santa Cruz Biotechnology | Santa Cruz: sc-59624 | 1:300 |
| Antibody | anti-myc 9E10 | Santa Cruz Biotechnology | Santa Cruz: sc-40 | 1:300 |
| Antibody | anti-tubulin | Sigma-Aldrich | Sigma: T6074 | 1:10,000 |
| Antibody | anti-nucleoporin-62 | BD Biosciences | BD Biosciences: 610498 | 1:1,000 |
| Antibody | anti-vimentin D21H3 | Cell Signaling | Cell Signaling: 5741 | 1:1,000; 1:100 |
| Antibody | anti-vimentin V9 | Sigma-Aldrich | Sigma-Aldrich: V6389 | 1:1,000 |
| Antibody | anti-GFP | Thermo Fisher Scientific | ThermoFisher: A11122 | |
| Antibody | Goat anti-rabbit IgG | Southern Biotechnology | Southern Biotech: 4030-05 | 1:5,000 |
| Antibody | goat anti-mouse IgG | Southern Biotechnology | Southern Biotech: 1030-05 | 1:5,000 |
| Antibody | goat anti-mouse k light chain | Southern Biotechnology | Southern Biotech: 1050-05 | 1:5,000 |
| Antibody | Goat anti-rabbit IgG (H+L) Alexa Fluor 594 conjugate | Thermo Fisher Scientific | Thermo Fisher Scientific: A-11012 | 1:5,000 |
| Antibody | goat anti-mouse IgG (H+L) Alexa Fluor 594 conjugate | Thermo Fisher Scientific | Thermo Fisher Scientific: A-11005 | 1:5,000 |
| Antibody | goat anti-rabbit IgG (H+L) IRDye 800CW conjugate | Li-Cor | Li-Cor: 925-32211 | 1:5,000 |

*Continued on next page*

*Continued*

| Reagent type (species) or resource | Designation | Source or reference | Identifiers | Additional information |
|---|---|---|---|---|
| Antibody | goat anti-mouse IgG (H+L) IRDye 800CW conjugate | Li-Cor | Li-Cor: 925-32210 | 1:5,000 |
| Recombinant DNA reagent | mEmerald-vimentin-N-18 | Addgene | Addgene: 54301 | |
| Recombinant DNA reagent | pLenti CMV/TO Hygro DEST | | Addgene plasmid: 17291 | |
| Recombinant DNA reagent | pLenti CMV Neo DEST (705-1) | | Addgene plasmid: 17392 | |
| Recombinant DNA reagent | mEmerald-vimentin-N-18/ pLenti6 CMV Neo | this paper | | |
| Recombinant DNA reagent | mEmerald-vimentin-N-18/ pLenti6 CMV Hygro | this paper | | |
| Chemical compound, drug | Thiamet-G | Duke Small Molecule Synthesis Facility | | |
| Chemical compound, drug | Ac45SGlcNAc | Benjamin Swarts, Central Michigan University | | |
| Chemical compound, drug | Advansta ECL | Advansta | | |
| Sequence-based reagent | vimentin sgRNA | Duke Functional Genomics Facility | | |
| Sequence-based reagent | vimentin sgRNA | Duke Functional Genomics Facility | | |
| Sequence-based reagent | vimentin sgRNA | Duke Functional Genomics Facility | | |
| Biological sample (virus) | Cas9 Lentivirus | Duke Functional Genomics Facility | | |
| Commercial assay or kit | pENTR Directional TOPO cloning kit | Thermo Fisher Scientific | Thermo Fisher Scientific: K240020 | |
| Commercial assay or kit | Gateway LR Clonase II Enzyme | Thermo Fisher Scientific | Thermo Fisher Scientific:11791100 | |

## Chemicals and enzymes

Thiamet-G was synthesized as described (*Yuzwa et al., 2008*) by the Duke Small Molecule Synthesis Facility (DSMSF). Ac$_4$5SGlcNAc was synthesized as described (*Gloster et al., 2011*) and was a gift of Benjamin Swarts, Central Michigan University. Ac$_3$GlcNDAz-1P(Ac-SATE)$_2$ was synthesized in-house or by the DSMSF as described (*Yu et al., 2012*). All other chemicals were purchased from Sigma unless otherwise noted. Lambda phosphatase was purchased from New England Biolabs.

## Mammalian cells and cell culture

Cells were maintained in Dulbecco's Modified Eagle's Medium (DMEM) supplemented with 10% fetal bovine serum (FBS), 100 units/ml penicillin and 100 µg/ml streptomycin and kept at 37°C with 5% $CO_2$. Cell lines were obtained from ATCC or the Duke Cell Culture Facility. In all cases, authenticity was verified using morphology, karyotyping, and PCR-based approaches (e.g., short tandem repeat profiling) and tested negative for *Mycoplasma* (PCR test) by the vendor at the time of purchase. HeLa and 293T cells were selected because they are well-established model systems for the human IF cytoskeleton and *Chlamydia* infection.

## GlcNDAz crosslinking

O-GlcNAc-mediated protein crosslinking was performed essentially as described (*Yu et al., 2012*). Briefly, Ac$_3$GlcNDAz-1P(Ac-SATE)$_2$ (GlcNDAz precursor) was added to a final concentration of 100 µM to the culture medium of 293T cells stably expressing AGX1(F383G) (*Yu et al., 2012*). DMSO served as the vehicle control treatment. Dishes were incubated in the dark for 24 hr, dosed again with GlcNDAz precursor, and incubated for an additional 24 hr.

To crosslink and harvest, dishes were placed on ice and washed carefully twice with 10 ml of cold phosphate-buffered saline (PBS) twice. With dishes still on ice, 4 ml of cold PBS were added, lids were removed, and plates were exposed to 365 nm UV light for 20 min. Then, cells were resuspended in PBS by scraping and/or pipetting, pelleted by centrifugation, and lysed in either 8 M urea for IB, or IP buffer (150 mM NaCl, 20 mM Tris pH 7.4, 1% Triton X-100, 0.1% SDS) for IPs.

## Immunoblotting (IB)

The concentrations of protein samples were obtained via bicinchoninic acid (BCA) assay according to the manufacturer's instructions (Thermo) and protein concentration was normalized across all samples within each experiment. One-third the volume of 4X SDS-PAGE loading buffer was added and the sample was heated to 95°C for 3 min (except in the case of samples prepared in 8 M urea). Samples were then loaded onto a polyacrylamide gel and run at 165 V.

Unless otherwise noted, IBs detection was performed by enhanced chemiluminescence (ECL). For ECL IBs, SDS-PAGE gels were transferred onto PVDF membrane using a BioRad semi-dry transfer system with transfer buffer (25 mM Tris, 192 mM glycine, 0.1% SDS, pH 8, 20% methanol) using standard methods. After transfer, membranes were blocked in 5% bovine serum albumin (BSA) in Tris-buffered saline with Tween (TBST; Tris-buffered saline, 0.1% Tween-20) overnight at 4°C with gentle rocking. Primary antibody dilutions were prepared in TBST+BSA and incubated overnight with gentle shaking at 4°C. The next day, membranes were washed three times with TBST and incubated with the appropriate secondary antibody for 1.5 hr at room temperature. Membranes were again washed three times with TBST, developed with Advansta ECL reagent, and exposed to film. The following primary antibodies were used: Anti-O-GlcNAc RL2 (Santa Cruz sc-59624), anti-myc 9E10 (Santa Cruz sc-40), anti-tubulin (Sigma T6074), anti-nucleoporin-62 (BD Biosciences 610498), anti-vimentin (clone D21H3, Cell Signaling 5741), anti-vimentin (clone V9, Sigma-Aldrich V6389), anti-phospho-(Ser/Thr) Akt substrate (Cell Signaling 59624), anti-GFP (ThermoFisher; A11122), MOMP (a gift from Ken Fields, University of Kentucky). The following horseradish peroxidase-conjugated secondary antibodies were used: Goat anti-rabbit IgG (Southern Biotech 4030–05), goat anti-mouse IgG (Southern Biotech 1030–05), goat anti-mouse κ light chain (Southern Biotech 1050–05).

For quantitative fluorescent IBs, samples were separated by SDS-PAGE, electroblotted onto nitrocellulose membranes, blocked and probed as described above. The following secondary antibodies were used: Goat anti-rabbit IgG (H + L) Alexa Fluor 594 conjugate (Thermo Fisher Scientific A-11012), goat anti-mouse IgG (H + L) Alexa Fluor 594 conjugate (Thermo Fisher Scientific A-11005), goat anti-rabbit IgG (H + L) IRDye 800CW conjugate (Li-Cor, 925–32211), goat anti-mouse IgG (H + L) IRDye 800CW conjugate (Li-Cor, 925–32210). Blots were washed in TBST and scanned and analyzed on a Li-Cor Odyssey imaging system.

## Plasmid construction

The sequence of human vimentin isoform 1 was amplified using PCR and the following primers: Forward: 5' cgcggatccgccaccatgtccaccaggtccgtg 3'; Reverse: 5' cgtctagattcaaggtcatcgtgatgctga 3'. The resulting PCR product and pcDNA3.1(+)/myc-His A vector (Invitrogen) were digested with BamHI and XbaI (New England Biolabs) and the vector treated with calf intestinal alkaline phosphatase (New England Biolabs) according to the manufacturer's instructions. PCR product and vector were mixed in a 3:1 molar ratio and incubated with T4 DNA ligase (New England Biolabs) in the provided buffer for 1 hr. This reaction was transformed into *E. coli* strain DH5α and the resulting colonies were cultured, miniprepped and sequenced using standard methods. The OGT-myc/6xHis and OGA-myc constructs have been described previously (*Boyce et al., 2011*).

The sequence of mEmerald-vimentin-N-18 (Addgene plasmid #54301) was amplified using primers complementary to vimentin and GFP (Forward: 5' caccgccaccatgtccaccaggtccgtg 3'; Reverse: 5' cgcaacgaattctcaatgtccaccaggtccgt). The resulting PCR product was cloned into a pENTR vector using the pENTR Directional TOPO cloning kit (Fisher Scientific #K240020) according to the manufacturer's instructions. This vector was used in a Gateway LR Clonase II Enzyme reaction, according to manufacturer's instructions (Invitrogen), with pLenti CMV Neo DEST (705-1) (Addgene plasmid #17392), or pLenti CMV/TO Hygro DEST (Addgene plasmid # 17291). Vimentin mutants were created from WT expression constructs via standard site-directed mutagenesis protocols using Agilent online primer design tools, oligonucleotides synthesized by IDT, and Phusion polymerase according to the manufacturer's instructions (Thermo Fisher). Mutagenesis was performed on vimentin-mEmerald in the pENTR vector prior to transfer to the appropriate destination vector. The integrity of all constructs was confirmed by Sanger sequencing through the entire open reading frame of each construct (Eton Bioscience).

## Transfections

30 µl of Trans-IT-293 transfection reagent (Mirus) was added to 750 µl of Opti-MEM medium (Life Technologies), vortexed and centrifuged briefly, and incubated at room temperature for 15 min. Then, 10 µg DNA was added to the mixture and vortexing and centrifuging were repeated, followed by an additional 15 min at room temperature. The entire reaction was added drop-wise to adherent cells.

## Immunoprecipitation (IP)

Samples were extracted in IP buffer (150 mM NaCl, 20 mM Tris pH 7.4, 1% Triton X-100, 0.1% SDS), diluted to 1 mg/ml total protein concentration, and 3 µg of antibody was added for each 1 mg of protein used. IPs were incubated at 4°C overnight with gentle agitation. The next day, washed protein-A/G agarose beads (Thermo) were added to the tubes and rotated at room temperature for 1.5 hr. The beads were then isolated by centrifuging at 600 g for 1 min and the supernatants were discarded. Beads were washed four times with 1 ml of IP buffer and eluted with three times the bead volume of elution buffer (8M urea, 150 mM NaCl; if the samples were to be further purified by nickel, this buffer also contained 10 mM imidazole).

## Differential extraction

Differential extraction was performed essentially as described (*Ridge et al., 2016*). Briefly, 293 T cells were grown to confluency, treated with GlcNDAz precursor and UV-crosslinked as above, and washed three times on the plate with PBS containing 2 mM $MgCl_2$ at room temperature. After removing PBS, attached cells were incubated in 1 ml of low-detergent buffer (10 mM MOPS pH 7, 10 mM $MgCl_2$, 1 mM EGTA, 0.15% Triton X-100, plus 7.5 µl saturated phenylmethane sulfonyl fluoride (PMSF) solution in ethanol, added fresh, in 1x PBS) for five minutes at room temperature with gentle agitation. The buffer was removed and cleared by benchtop centrifugation, and the resulting supernatant was reserved as the soluble cell fraction. Plates were then incubated on ice in 1 ml of ice-cold high-detergent buffer (10 mM MOPS pH 7, 10 mM MgCl2, 1% Triton X-100, plus 7.5 µl saturated PMSF solution and 50 mg Benzonase, added fresh, in 1x PBS) for 3 min. Then, 250 µl of ice-cold 5 M NaCl was added, cells were resuspended by pipetting, and samples were cleared by centrifugation. The resulting supernatant was saved as the cytoskeletal fraction, including intermediate vimentin assembly states (*Ridge et al., 2016*). Finally, the pellet was resuspended by pipetting in 250 µl of 8 M urea in PBS to solubilize the remaining material (e.g., assembled IFs) (*Ridge et al., 2016*).

## Buffer exchange

Zeba spin desalting columns (Thermo Fisher, 89890) were washed with 1 ml of IP buffer and centrifuged at 1000 g for 2 min three times, according to the manufacturer's instructions. Samples were applied to the center of the column and 40 µl of IP buffer was applied after sample absorption into the resin. The column was centrifuged for 2 min at 1000 g and sample was collected in 1.5 ml centrifuge tubes.

## Mass spectrometry-based proteomics

293T cells stably expressing AGX1(F383G) (*Yu et al., 2012*) were transfected with a vimentin-myc-6xHis construct, treated with GlcNDAz precursor, and UV-crosslinked as above. Crosslinked and uncrosslinked vimentin-myc-6xHis was isolated first through anti-myc IP, as above. Samples were eluted from the protein A/G beads in 8 M urea, 150 mM NaCl and 10 mM imidazole in PBS and incubated with nickel-NTA resin (Qiagen) rotating at room temperature for 2 hr. The resin was washed three times with the same buffer and eluted with 8 M urea, 150 mM NaCl and 250 mM imidazole in PBS for 15 min. Eluents were separated by SDS-PAGE and stained with InstantBlue gel stain (Thermo Fisher). Bands corresponding to crosslinks were excised by hand, and analyzed by in-gel digest and MS/MS proteomics by the Duke Proteomics and Metabolomics Shared Resource. For more details, please see https://genome.duke.edu/cores-and-services/proteomics-and-metabolomics/protein-characterization

## GalNAz labeling and click reactions

GalNAz labeling was performed essentially as described (*Boyce et al., 2011*; *Palaniappan et al., 2013*; *Chen et al., 2017*). Briefly, 293T cells were incubated with DMSO vehicle or 100 µM Ac$_4$Gal-NAz for 24 hr. To harvest, cells were washed twice with PBS and resuspended in click lysis buffer (1% Triton X-100, 1% SDS, 150 mM NaCl, 20 mM Tris pH 7.4, 5 µM PUGNAc, protease inhibitor cocktail). After sonication and centrifugation, protein concentration was measured via BCA assay.

2.5 mg of protein were used in 875 µl total reaction volume for each click reaction. To these reactions were added 5 mM sodium ascorbate, 25 µM alkyne-Cy5 probe (*Palaniappan et al., 2013*), 100 µM TBTA, and 1 mM CuSO$_4$. Samples were incubated in the dark at room temperature for 1 hr with gentle agitation and quenched with 10 mM EDTA. Samples were then diluted with 1% Triton X-100, 150 mM NaCl, 20 mM Tris pH 7.4, 1 mM EDTA, 5 µM PUGNAc and protease inhibitors to bring SDS concentration to 0.1%. 6.75 µg anti-myc (9E10) antibody were added to each sample and IPs were incubated at 4°C overnight with gentle agitation. The next day, washed protein-A/G agarose beads (Thermo) were added to the tubes and rotated at room temperature for 1 hr. The beads were then isolated by centrifuging at 600 g for 1 min and the supernatants were discarded. Beads were washed four times with 1 ml of IP buffer, eluted with three times the bead volume of SDS-PAGE loading buffer, and boiled for 5 min at 95°C. Samples were analyzed by SDS-PAGE and fluorescence scanning on a Li-Cor Odyssey imaging system.

## Generation and validation of vimentin$^{-/-}$ cells by CRISPR/Cas9 deletion

Three single guide RNA (sgRNA) sequences targeting the human vimentin locus were designed and validated via the Surveyor assay (*Ran et al., 2013*) by the Duke Functional Genomics facility: Vim-1: 5' GGACGAGGACACGGACCTGG 3'; Vim-2: 5' CATCCTGCGGTAGGAGGACG 3'; Vim-3: 5' GGACACGGACCTGGTGGACA 3'. An sgRNA targeting the AAVS1 'safe harbor' locus (*Sadelain et al., 2011*) was used as a control.

HeLa or 293T cells were added to a 6-well plate at ~40–60% confluency and allowed to attach for 24 hr. Then, cells were infected by adding 50 µl of a Cas9 lentivirus (Addgene plasmid #52961), produced by the Duke Functional Genomics Facility, in DMEM plus 8 µg/ml polybrene drop-wise to each well. Plates were centrifuged at 700 g for 30 min and incubated under standard conditions for 24 hr. Then, the medium was changed to DMEM without polybrene and the cells were allowed to recover for 3 days. Cells were selected and maintained with blasticidin (5 µg/ml for 293T, 3 µg/ml for HeLa) for several passages before infection with sgRNA viruses.

HeLa or 293T cells stably transduced with lentiviral Cas9 were added to a 6-well plate at ~40–60% confluency and allowed to attach for 24 hr. Then, cells were infected with sgRNA-expressing lentivirus produced by the Duke Functional Genomics Facility in DMEM plus 8 µg/ml polybrene, and 50 µl of virus was added drop-wise to each well. Plates were centrifuged at 700 g for 30 min and incubated under standard conditions for 24 hr. Then, the medium was changed to DMEM without polybrene and the cells were allowed to recover for 3 days. Cells were selected and maintained with puromycin (0.5 µg/ml for 293T, 1.5 µg/ml for HeLa) for several passages. Single cells were sorted into 96-well plates by a DiVa fluorescence-activated cell sorter (BD Biosciences) at the Duke Cancer Institute Flow Cytometry Shared Resource (DCI FCSR) to obtain individual clones. Clones were screened for successful vimentin deletion both by IP/IB with a monoclonal vimentin antibody (V9, Sigma) and quantitative RT-PCR (qPCR).

For qPCR, cellular mRNA was extracted with an RNeasy kit according to the manufacturer's instructions (Qiagen). RNA was used to generate cDNA using SuperScript II reverse transcriptase according to the manufacturer's instructions (Thermo Fisher #18064014). cDNA was diluted five-fold, and triplicate reactions were performed using Sir Master Mix (Life Tech-Power Sybr no.4367659) according to manufacturer's instructions. Reactions were performed on a StepOnePlus Real-Time PCR system (Applied Biosystems).

The following qPCR primers and cycling conditions were used:

## Vimentin

Forward: 5'-AGTGTGGCTGCCAAGAACCT 3'
Reverse: 5'-GAGGGACTGCACCTGTCTCC 3'

## Actin

Forward: 5'-CACTCTTCCAGCCTTCCTTC 3'
    Reverse: 5'-GGATGTCCACGTCACACTTC 3'

| Step | Temperature (°C) | Time (minutes) | Cycles |
|------|------------------|----------------|--------|
| Initial denaturation | 95 | 10 | 1 |
| Denaturation | 95 | 0.15 | 40 |
| Annealing | 60 | 0.3 | |
| Extension | 72 | 0.3 | |
| Final extension | 72 | 10 | 1 |
| Storage | 4 | ∞ | - |

## Creation of vimentin-mEmerald lentivirus and reconstitution of vimentin$^{-/-}$ cells

$1.2 \times 10^6$ 293 T cells were plated at ~60–70% confluency in a 6 cm dish 24 hr prior to transfection. Cells were transfected with:

- 1 µg vimentin expression construct (lentivector, produced as described above)
- 900 ng psPAX2 (packaging plasmid; Addgene plasmid # 12260)
- 100 ng pMD2.G (VSV-G/ENV plasmid; Addgene plasmid # 12259)

12–18 hr post-transfection, the medium was changed to 4 ml of DMEM with 30% FBS. At 48 hr post-transfection, the medium containing the virus was harvested and replaced with fresh DMEM. At 72 hr post-transfection, the virus-containing medium was again harvested, combined with the previous medium, and filtered through a 0.45 µm PVDF filter. Filtered supernatants were then used to infect target cells as described above. HeLa cells were selected and passaged in 500 µg/ml G418, and 293T cells with 100 µg/ml hygromycin. After trypsinizing, cells were resuspended in serum-free DMEM (+penicillin/streptomycin) and passed through a sterile 30 µm filter (Sysmex CellTrics, 04-004-2326). Cells were sorted on a DiVa fluorescence-activated cell sorter (BD Biosciences) at the DCI FCSR to obtain the top third of highest-expressing cells, as judged by GFP fluorescence (vimentin-mEmerald signal).

## Filament imaging

Imaging of vimentin-mEmerald was performed using a confocal laser scanning microscope (LSM 880; Zeiss) equipped with an automatic stage, Airyscan detector (Hamamatsu) and diode (405 nm), argon ion (488 nm), double solid-state (561 nm), and helium-neon (633 nm) lasers. Images were acquired using a 60x/1.4 NA oil objective (Zeiss) and deconvolved using automatic Airyscan Processing in the Zen Software (Zeiss).

## Immunofluorescence

Cells were rinsed twice with 37°C PBS and fixed with 1% formaldehyde (Sigma) in PBS for 10 min. Cells were permeabilized with PBS containing 0.1% Triton X-100 (Sigma) for 10 min and blocked with TBS containing 5% BSA (Equitech-Bio) and 0.1% Triton X-100. A mouse antibody against vimentin (clone V9, Santa Cruz Biotech sc-57678) was diluted in TBS containing 5% BSA and 0.1% Triton X-100 and incubated with the samples overnight at 4°C. Cells were washed three times with PBS and then incubated with a goat anti-rabbit (H + L) Alexa Fluor 647-conjugated secondary antibody (Thermo Fisher) diluted in TBST containing 5% BSA for 1 hr at room temperature. Coverslips were washed five times with PBS and mounted on slides using ProLong Diamond anti-fade mounting medium with DAPI (Thermo Fisher). Cells were imaged using a confocal laser scanning microscope (LSM 880; Zeiss) equipped with an automatic stage, Airyscan detector (Hamamatsu) and diode (405 nm), argon ion (488 nm), double solid-state (561 nm), and helium-neon (633 nm) lasers. Images were acquired using a 60x/1.4 NA oil objective (Zeiss) and deconvolved using automatic Airyscan Processing in the Zen Software (Zeiss).

## Transwell migration assays

Transwell migration assays were performed essentially as described (*Justus et al., 2014*). Briefly, cells were plated at approximately 70% confluency and allowed to attach for 24 hr. Then, the medium was removed and replaced with DMEM containing penicillin/streptomycin but lacking FBS for 72 hr. 6.5 mm Transwell plates with 8.0 µm pore polycarbonate membrane inserts were collagen-coated by incubating individual inserts in 50 µg/ml collagen solution from bovine skin (Sigma-Aldrich, C4243-20ML) for 1 hr at 37°C, UV-sterilized in a biosafety cabinet, and re-hydrated with FBS-free DMEM for 1 hr. FBS-starved cells were trypsinized and counted, and 30,000 cells per replicate were added to each insert with either FBS-containing or FBS-free DMEM on the opposite side. Cells were permitted to migrate for 24 hr under standard culture conditions. The assay was stopped by fixing cells in ice-cold methanol for 10 min at −20°C. Then, inserts were stained with crystal violet solution (30% methanol, 0.1% crystal violet in PBS) overnight. After staining, inserts were washed three times in PBS and the non-migrated cells were gently removed with a cotton swab. Four non-overlapping fields of view per insert were imaged with a 10x objective of a Nikon TE200 inverted microscope. Cells were counted manually using Fiji (NIH).

## Statistical analysis

For filament morphology quantification, the number of puncta- and filament-containing cells was normalized as a percent of total cells counted and analyzed by ANOVA followed by pairwise t-tests, with $p < 0.05$ considered significant. Transwell migration assays were normalized to percent of control (i.e., WT vimentin, vehicle-treated, serum-stimulated) migration and analyzed by ANOVA followed by pairwise t-tests, with $p < 0.05$ considered significant.

## *Chlamydia* strains and elementary body preparations

*Chlamydia trachomatis* serotype LGV-L2, strain 434/Bu (CTL2) was propagated in Vero cells. Infectious elementary bodies (EBs) were derived from Vero-infected cells at 44 hr post-infection (hpi). Infected cells were rinsed twice with PBS, lysed in water for 10 min, and diluted in buffer (7.2 mM $K_2HPO_4$, 3.8 mM $KH_2PO_4$, 218 mM sucrose, 4.9 mM L-glutamic acid, pH 7.4). Cell lysates were subsequently sonicated and stored at −80°C.

## *Chlamydia* infections

HeLa cells were seeded at a density of $5 \times 10^4$ cells/well on glass coverslips (Bellco Glass, Inc.). Coverslips were pre-coated with 30 µg/ml type I collagen (Thermo Fisher) in 20 mM acetic acid (Spectrum) for 5 min and rinsed twice with medium. Cells were maintained in medium containing 0.5 mg/ml G418 and rinsed three times with medium lacking G418 just prior to infections. CTL2 EBs were added at a multiplicity of infection of three and infections were synchronized by centrifugation (1000 g, 20 min) at 10°C. The medium was replaced and infected cells were cultured under standard conditions for 30 hr. To inhibit OGT or OGA activity, the medium of infected cells was replaced with medium containing DMSO (vehicle), 50 µM 5SGlcNAc, or 50 µM Thiamet-G at 10 hpi.

## Immunofluorescence in *Chlamydia* infection experiments

HeLa cells were rinsed twice with warm PBS and fixed with 3.7% formaldehyde in PBS for 20 min. Cells were quenched with 0.25% $NH_4Cl$, permeabilized with PBS containing 0.1% Triton X-100 for 10 min, and blocked with PBS containing 2% BSA and 0.1% Triton X-100. A mouse antibody against MOMP (Santa Cruz, sc-57678) and a rabbit antibody against cap1 (A. Subtil, Institut Pasteur) were diluted in PBS containing 2% BSA and 0.1% Triton X-100. The secondary antibodies goat-anti-mouse (H + L) Alexa Fluor 647 (Thermo Fisher) and goat-anti-rabbit (H + L) Alexa Fluor 555 (Thermo Fisher) were diluted in PBS containing 2% BSA and 0.1% Triton X-100. Coverslips were washed five times with PBS and mounted on slides using Fluorsave mounting media (CalBiochem).

Cells were imaged using a confocal laser scanning microscope (LSM 880; Zeiss equipped with an automatic stage, Airyscan detector (Hamamatsu) and diode (405 nm), argon ion (488 nm), double solid-state (561 nm), and helium-neon (633 nm) lasers. Images were acquired using a 60x/1.4 NA oil objective (Zeiss) and deconvolved using automatic Airyscan Processing in the Zen Software (Zeiss).

## Analysis of *Chlamydia* infection experiment images

To quantify *Chlamydia* inclusion size, images were imported into ImageJ (NIH) and converted to 8-bit TIFF and binary image files to demarcate individual inclusions. The area of each cap1-positive inclusion and the number of extra-inclusion bacteria were exported and plotted in the R software. Datasets were analyzed in R using Levene's Test to assess equal variance, followed by either a Student's t-test or Welch's t-test, with $p < 0.05$ considered significant.

## Vimentin cleavage

Vimentin$^{-/-}$ HeLa cells stably transduced with empty vector or WT, S49A or Y117L vimentin-mEmerald were mock-infected or infected with CTL2 *Chlamydia* (MOI = 0.5) for 30 hr, washed twice with cold PBS, incubated in ice-cold buffer (50 mM Tris pH 7.4, 150 mM NaCl, 1 mM EDTA, 1 mM PMSF, 1% Triton X-100, and protease inhibitors (Roche)) for 30 min and then sonicated on ice for 10 s. Cell lysates were cleared by centrifugation at 8000 rpm for five minutes at 4°C. Supernatants were diluted in SDS-PAGE sample buffer and heated to 95°C for five minutes. Equal volumes of sample were analyzed by IB. Rabbit antibodies against GFP (ThermoFisher A11122) and MOMP (a gift from Ken Fields), a mouse antibody against α-tubulin (Sigma-Aldrich, Clone B-5-1-2), and secondary anti-rabbit and anti-mouse antibodies (Li-Cor Biosciences) conjugated to infrared dye were diluted in PBS containing 5% nonfat milk (weight/volume) and 0.1% Tween-20, and sequentially incubated on the membrane prior to scanning with the Odyssey imaging system (Li-Cor Biosciences).

## Transparent reporting

We have adhered to the definition of 'biological replicates' outlined by *Blainey et al. (2014)* and paraphrased as independent, parallel measurements of biologically distinct samples to capture biological variation. By contrast, technical replicates are repeated measures of the same biological sample to determine the noise associated with experimental procedures or instruments (*Blainey et al., 2014*). Individual experiments shown in the figures are representative of at least three biological replicates. Statistical standards and tests for particular experiments are detailed in the appropriate subsections of the Materials and Methods and figure legends. For more information, please see the *eLife* Transparent Reporting form associated with this work.

# Acknowledgements

We thank Ken Fields (University of Kentucky) for anti-MOMP antibody, Jennifer Kohler (University of Texas Southwestern Medical Center) for plasmids, Benjamin Swarts (Central Michigan University) for Ac$_4$5SGlcNAc, Agathe Subtil (Institut Pasteur) for anti-cap1 antibody, Brittany Bisnett for advice on statistical analysis, and members of the Boyce Lab for helpful suggestions. This work was supported by a Rita Allen Foundation Scholar Award and NIH grant 1R01GM118847-01 to MB, NIH grants 1R01AI123083 and AI107951 to RHV, and NIH grant P41-EB002025 to the Computer Integrated Systems for Microscopy and Manipulation at the University of North Carolina at Chapel Hill, supporting ETO.

# Additional information

### Funding

| Funder | Grant reference number | Author |
| --- | --- | --- |
| National Institute of Biomedical Imaging and Bioengineering | P41-EB002025 | E Timothy O'Brien III |
| National Institute of Allergy and Infectious Diseases | 1R01AI123083 | Raphael H Valdivia |
| National Institute of Allergy and Infectious Diseases | 1R01AI107951 | Raphael H Valdivia |
| National Institute of General Medical Sciences | 1R01GM118847-01 | Michael Boyce |

| Rita Allen Foundation | Scholar Award | Michael Boyce |

The funders had no role in study design, data collection and interpretation, or the decision to submit the work for publication.

## Author contributions

Heather J Tarbet, Conceptualization, Resources, Supervision, Funding acquisition, Investigation, Methodology, Writing—original draft, Project administration, Writing—review and editing; Lee Dolat, Conceptualization, Data curation, Formal analysis, Validation, Investigation, Visualization, Methodology, Writing—original draft, Writing—review and editing; Timothy J Smith, Conceptualization, Resources, Data curation, Formal analysis, Investigation, Methodology, Writing—review and editing; Brett M Condon, Investigation, Methodology; E Timothy O'Brien III, Software, Formal analysis, Validation; Raphael H Valdivia, Resources, Software, Formal analysis, Writing—review and editing; Michael Boyce, Conceptualization, Resources, Formal analysis, Supervision, Funding acquisition, Investigation, Methodology, Project administration, Writing—review and editing

## Author ORCIDs

Michael Boyce (iD) http://orcid.org/0000-0002-2729-4876

## Decision letter and Author response

Decision letter https://doi.org/10.7554/eLife.31807.024
Author response https://doi.org/10.7554/eLife.31807.025

## Additional files

### Supplementary files

• Transparent reporting form
DOI: https://doi.org/10.7554/eLife.31807.022

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
