## [Decision Letter]

Thank you for submitting your article "Site-specific glycosylation regulates the form and function of the intermediate filament cytoskeleton" for consideration by *eLife*. Your article has been reviewed by three peer reviewers, and the evaluation has been overseen by a Reviewing Editor and Michael Marletta as the Senior Editor. The following individuals involved in review of your submission have agreed to reveal their identity: John A Hanover (Reviewer #1); Chad Slawson (Reviewer #2).

The reviewers have discussed the reviews with one another and the Reviewing Editor has drafted this decision to help you prepare a revised submission. Although the reviewers are interested in your findings, there is some considerable concern with some aspects of the data (e.g. point 4).

Summary:

This is an interesting paper documenting an approach to understand the role of O-GlcNAcylation on IF structure and function. The authors demonstrate that O-GlcNAyclation of vimentin is important for normal function of the IF protein and that S49 on vimentin is the most critical O-GlcNAc site on the molecule. The studies with Chlamydia further extend the significance of the paper. The cross-linking approach uncovers interactions normally not accessible by other techniques.

Essential revisions:

1) An important aspect of the conclusions made in this study related to the GlcNDAz crosslinking assay. I have several concerns that will need some testing in some cases and comments in others: i) Why is there significant variability is the relative stoichiometry of the crosslinked species vs. the vimentin monomer? ii) Related to point 1-i, do different anti-vimentin antibodies provide similar crosslinking profiles and can this potentially tell us something about the location of the crosslink (e.g., see the extent of myc vs. vimentin crosslinks in Figure 1—figure supplement 1?. iii) The crosslinking efficiency likely depends on the conformation of the involved proteins. What is the crosslinking efficiency of vimentin in the presence or absence of 5SGlnNAc or Thiamet? iv) What happens to vimentin solubility in the context of different crosslinking conditions?

2) Protein expression of the various constructs needs to be shown for at least some of the experiments where expression levels can alter the conclusion. This is particularly relevant for the cell migration effects shown in Figure 4. Expression of vimentin clearly enhances serum induced cell migration but the phenotype of the S49A and Y117L may be related to vimentin levels and any potential effect on the mutants on cell growth,

3) If S49 glycosylation is important for baseline filament organization, one prediction is that its glycosylation stoichiometry would be expected to be high. It would be important to compare the glycosylation of vimentin WT vs. some of the other glycol-mutants shown in Figure 3. This can be done using in vitro galactosylation.

4) Although the Plasmid Construction section of the Materials and methods indicates that the constructs were '..miniprepped and sequenced…" it would be important that the entire clones are sequenced in full (i.e., not only the inserts). May be this was done already but I am a bit surprised that all the mutants shown in Figure 3 (except for the T33A mutant) had such a profound effect on vimentin organization.

5) Figure 5: the effect of 5SGlcNAc on inclusion area in HeLa cells that lack vimentin is not significant. What about the number of extra-inclusion bacteria?

6) It is not surprising that the inclusion size is smaller for the two vimentin mutants (S49A and Y117L, Figure 5). Disruption of the vimentin cytoskeleton is predicted to interfere with any cellular process that involves vimentin reorganization. A previous report suggested the presence of a chlamydia-secreted protease that disrupts the keratin cytoskeleton (F Dong et al., Infect Immun 2004). It is possible that the vimentin mutations have altered susceptibility to degradation vs. WT vimentin. It would be important to tone down the importance of glycosylation in this process.

7) How conserved is the S49-containing motif among other intermediate filament proteins. If conserved, are there known mutations in this residue in other intermediate filament proteins.

8) Did the authors use OGA to cleave GlcNDAz in vitro (and see things go away) As a side note, GlcNDAz isn't turned over well in the cell and accumulates – did they note any positive/negative consequences of that in their experiments? Why cycling? Not presence of?

9) Results, ninth paragraph: do cells in which OGT/OGA are perturbed have same characteristics? This would suggest it's cycling (esp. if it's present w/ OGA perturbation) rather than the addition of OGlcNAc (and just OGT disruption).

10) “Taken together, these results indicate that O-GlcNAcylation of vimentin, especially on S49, is required for optimal serum-stimulated cell migration.” Is that the case? If so, does it get better with TMG treatment or siRNA orf OGA?

11) Results, end of eleventh paragraph: is it possible that the 5SGlcNAc influences the chlamydia itself rather than just the host cell? What if it is treated with 5SGlcNAc (even if it's not known to have OGT, it might have an effect?)

12) Results, end of last paragraph: did increased o-glcNAc improve response to chlamydia?

13) Figure 1: were Figure 1 from the urea extraction method? This doesn't have to be chronological, but it seems important to note given that you only see it in that fraction…

14) Figure 2 would mutations of these expect to change the folding for any reason other than O-GlcNAcylation changes? There seems no independent analysis of vimentin abundance and stability.

15) Figure 3: what is being compared? WT to the other samples in the t-test or just the filaments/punctae within a sample? All samples? Statistics are unclear.

16) Figure 4: here it is best to argue that residues subject to glycosylation are important for cell migration. Formally, it is difficult to tease out the role of O-GlcNAc at these sites since both the GlcNAc and the backbone is altered. It’s also hard to tell what's being noted as statistically different here. I presume it's the difference between WT and the other three when serum stimulated? This should be better described either in the main text or in the legend. (i.e., vector is the -/- cell line without being reconstituted?).

17) Were the cells +5SGlcNAc serum stimulated? Are these data the same as the samples on the left just with additional datapoints of the 5SGlcNAc? Consider just condensing to one graph or noting that. If the statistics become difficult and that's why it's split out, it should be easy to note in a table or graphically when "NS" is taken out. Also, for WT serum stimulated, are there error bars?

18) Figure 5 legend: -/- is what is indicated I think?

---

## [Author Response]

Essential revisions:1) An important aspect of the conclusions made in this study related to the GlcNDAz crosslinking assay.

We agree with the reviewer that GlcNDAz crosslinking is a useful method for characterizing O-GlcNAc-mediated protein-protein interactions, but would also benefit from additional control experiments and clarifications in our manuscript.

I have several concerns that will need some testing in some cases and comments in others: i) Why is there significant variability is the relative stoichiometry of the crosslinked species vs. the vimentin monomer?

Some variability in the extent of GlcNDAz-mediated vimentin crosslinking (i.e., crosslinked:uncrosslinked ratio) is likely due to cell type dependence. For example, the crosslinking in Figure 1 was performed in HeLa cells, whereas Figure 1 are from 293T cells. In addition, there is some intrinsic variability in the efficiency of each of the several steps in the GlcNDAz protocol, including the biosynthesis of UDP-GlcNDAz, the competition between UDP-GlcNDAz and abundant endogenous UDP-GlcNAc, the glycosylation of vimentin by OGT, the successful activation of the O-GlcNDAz diazirine by UV irradiation, and the productive insertion of the resulting carbene to form a covalent vimentin-vimentin crosslink (as opposed to the unproductive quenching of the carbene on a small molecule in the cytoplasm, for example)^1,2^. Since each of these steps is less than 100% efficient and subject to stochastic variability and/or cell type dependence, the final crosslinking results also vary somewhat. Importantly, however, these factors also mean that the crosslinking we observe almost certainly represents an underestimate of the extent of O-GlcNAc-mediated interactions among vimentin molecules, underlining their prevalence. In addition, the qualitative pattern of vimentin GlcNDAz crosslinking is highly reproducible across experiments and cell types (e.g., Figure 1, Figure 2, Figure 1—figure supplements 1, 7), indicating that it reports on specific, conserved interactions. We have addressed these considerations in the fifth paragraph of the Results section.

ii) Related to point 1-i, do different anti-vimentin antibodies provide similar crosslinking profiles and can this potentially tell us something about the location of the crosslink (e.g., see the extent of myc vs. vimentin crosslinks in Figure 1—figure supplement 1?

We have used two different commercial anti-vimentin monoclonal antibodies, D21H3 and V9, as well as the 9E10 anti-myc monoclonal antibody to detect GlcNDAz crosslinks of vimentin in several experiments (e.g., Figure 1, Figure 1—figure supplement 1 and 7 with D21H3, Figure 1—figure supplement 1 and Figure 1—figure supplement 2 with V9, and Figure 2 with 9E10). In the revised manuscript, we have stated which monoclonal was used in all applicable figure legends. We do not observe obvious differences in crosslinking when different antibodies are used (e.g., compare Figure 1, performed with D21H3, with Figure 1—figure supplement 1 and Figure 1—figure supplement 2, performed with V9). We agree with the reviewer that the site(s) of the vimentin crosslinks is of great interest, because knowing the site of carbene insertion into a second vimentin molecule might reveal a specific, O-GlcNAc-interacting motif. However, because, immunoblot (IB) analysis of GlcNDAz crosslinks is uniform across all antibodies used, this assay does not provide new information on that question. We have addressed these points to the fifth paragraph of the Discussion.

iii) The crosslinking efficiency likely depends on the conformation of the involved proteins. What is the crosslinking efficiency of vimentin in the presence or absence of 5SGlnNAc or Thiamet?

To address the reviewer’s question, we have now performed GlcNDAz crosslinking in DMSO-, 5SGlcNAc- and Thiamet-G-treated cells (Figure 1—figure supplement 1). As expected, 5SGlcNAc treatment reduces GlcNDAz crosslinking of vimentin. In contrast, Thiamet-G treatment has little effect on crosslinking, likely because basal levels of vimentin glycosylation are high under physiological OGT expression conditions. In addition, we agree with the reviewer that the conformation of vimentin likely affects the efficiency and molecular topology of GlcNDAz crosslinks. All glycosylation sites that we characterized lie within the vimentin head domain, which is thought to be unstructured^3-8^. Because we observe reproducible, discrete crosslink products (cf. Figure 1 and Figure 2, Figure 1—figure supplements 1, 7), our results raise the possibility that the head domain may adopt a specific conformation through O-GlcNAc-mediated interactions in intermediate filaments (IF) in vivo. We have addressed these considerations in the Discussion section of the revised manuscript (fifth paragraph).

iv) What happens to vimentin solubility in the context of different crosslinking conditions?

As the reviewer notes, we extracted crosslinked vimentin samples into different buffers depending on the experimental need. For example, standard IBs were usually performed on crosslinked samples prepared in denaturing urea buffer (e.g., Figure 1), whereas experiments involving immunoprecipitations (IPs) were performed with samples made in a standard, non-denaturing IP buffer (e.g., Figure 1—figure supplement 1). As the reviewer anticipates, it is important to confirm that vimentin crosslinks are soluble in both buffers, because vimentin crosslinking could not be reliably detected and compared if GlcNDAz crosslinking itself renders vimentin insoluble in those buffers. To address this point, we have now performed a control experiment wherein we performed the differential extraction assay^9^ and then exchanged the crosslinks from the denaturing urea buffer into non-denaturing buffer (Figure 1—figure supplement 2. This result demonstrates that the vimentin crosslinks are equally soluble in both cases, confirming the validity of comparing crosslink IBs performed with different buffers across experiments.

2) Protein expression of the various constructs needs to be shown for at least some of the experiments where expression levels can alter the conclusion. This is particularly relevant for the cell migration effects shown in Figure 4. Expression of vimentin clearly enhances serum induced cell migration but the phenotype of the S49A and Y117L may be related to vimentin levels and any potential effect on the mutants on cell growth.

We agree with the reviewer that uniform expression of different wild type and mutant vimentin constructs in our reconstituted cells is critical. To confirm even expression, we used fluorescence-activated cell sorting to normalize the vimentin-mEmerald expression in all reconstituted HeLa and 293T cell lines. In the revised manuscript, we have updated the Materials and methods text to explain this fact, and have included a new figure from the flow cytometry data, demonstrating even expression of the various vimentin-mEmerald expression across these cell lines (Figure 3—figure supplement 2). These flow cytometry data are consistent with the microscopy (Figure 3 and Figure 3—figure supplement 4A) and quantitative IB (Figure 3—figure supplement 2) data, which confirm uniform vimentin-mEmerald expression across cell lines at a level comparable to that of endogenous vimentin. Based on these controls, we believe the phenotypes we observe in our IF morphology, cell migration and *Chlamydia* infection assays are due to differences in vimentin genotype, and not differences in vimentin expression levels. We have revised the seventh paragraph of the Results section to clarify these points.

3) If S49 glycosylation is important for baseline filament organization, one prediction is that its glycosylation stoichiometry would be expected to be high. It would be important to compare the glycosylation of vimentin WT vs. some of the other glycol-mutants shown in Figure 3. This can be done using in vitro galactosylation.

We agree with the reviewer that the stoichiometry of vimentin S49 glycosylation is an interesting and important question. We attempted a galactosyltransferase assay, as the reviewer suggested, but observed poor labeling in all samples, including cell lysates and commercial positive control proteins (α-crystallin) (not shown). As an alternative approach, we performed metabolic labeling with *N*-azidoacetylgalactosamine (GalNAz) to quantify O-GlcNAcylation^10-12^. We have previously demonstrated that GalNAz labeling and subsequent bioorthogonal chemical ligation permit the detection of authentic O-GlcNAcylation in live human cells^10-12^. We GalNAz-labeled cells transfected with vector only or wild type or S49A vimentin constructs, ligated an alkyne-Cy5 probe to O-GlcNAz moieties, IP-ed vimentin and analyzed Cy5 signal to quantify the extent of glycosylation (Figure 2—figure supplement 2). In this experiment, the Cy5 signal of the S49A mutant was approximately 80% of the wild type, suggesting that S49 glycosylation accounts for about one-fifth of total vimentin glycosylation under these conditions. We believe these results are consistent with our model that significant levels of S49 glycosylation promote IF formation or stability in live cells (Figure 3, Figure 3—figure supplement 4). These data are discussed in the sixth paragraph of the Results section.

4) Although the Plasmid Construction section of the Materials and methods indicates that the constructs were '..miniprepped and sequenced…" it would be important that the entire clones are sequenced in full (i.e., not only the inserts). May be this was done already but I am a bit surprised that all the mutants shown in Figure 3 (except for the T33A mutant) had such a profound effect on vimentin organization.

We agree with the reviewer that the IF morphology and functional phenotypes we observe in vimentin mutants are striking, supporting the importance of vimentin glycosylation in vivo. In prior e-mail correspondence with Dr. Marletta and the *eLife* staff, we addressed the question of sequencing our vimentin constructs raised in this comment. Briefly, we sequence-verified the entire open reading frames of all vimentin constructs used in our work to confirm their integrity and to rule out the possibility of any adventitious mutations. Importantly, site-directed mutagenesis was performed in a donor vector, and then wild type or mutant vimentin open reading frames were transferred to a destination vector via standard topoisomerase-based cloning methods. This workflow eliminates the possibility that the mutagenesis procedure introduced undetected mutations in the backbone vector, since the destination vectors were never subjected to site-directed mutagenesis. We have revised the “Plasmid Construction” subsection of the Materials and methods to explain this point.

5) Figure 5: the effect of 5SGlcNAc on inclusion area in HeLa cells that lack vimentin is not significant. What about the number of extra-inclusion bacteria?

We agree with the reviewer that it would be useful to determine the effects of 5SGlcNAc treatment on extra-inclusion *Chlamydia* in infected vimentin^-/-^ HeLa cells. We performed this experiment and now show that OGT inhibition has no effect on the number of extra-inclusion bacteria in cells lacking vimentin (Figure 5, bottom panel, and Results, last paragraph). This result is consistent with our model that vimentin glycosylation sites and OGT activity are both required for IF remodeling by *Chlamydia*.

6) It is not surprising that the inclusion size is smaller for the two vimentin mutants (S49A and Y117L, Figure 5). Disruption of the vimentin cytoskeleton is predicted to interfere with any cellular process that involves vimentin reorganization. A previous report suggested the presence of a chlamydia-secreted protease that disrupts the keratin cytoskeleton (F Dong et al., Infect Immun 2004). It is possible that the vimentin mutations have altered susceptibility to degradation vs. WT vimentin. It would be important to tone down the importance of glycosylation in this process.

As the reviewer notes, the *Chlamydia*-encoded CPAF protease cleaves host IF proteins^13^. Indeed, we have previously shown that vimentin in particular is a CPAF substrate during infection^14-17^. We agree with the reviewer that the effects of vimentin glycosylation on vimentin cleavage during infection are an important question. We now provide IBs demonstrating that the infection-dependent cleavage of vimentin is unaffected in the S49A and Y117L mutants (Figure 5—figure supplement 9). The observations that the S49A mutation does not affect vimentin cleavage by CPAF (Figure 5—figure supplement 9) but still prevents IF remodeling (Figure 5) support the importance of vimentin glycosylation sites and OGT activity during *Chlamydia* infection. We also agree with the reviewer that any generic disruption in the vimentin cytoskeleton could interfere with IF reorganization during infection. In this regard, as the reviewer notes, the phenotypes we observe in the Y117L mutant are not surprising, since Y117L cannot assemble beyond the unit-length filament stage^18-21^. This property makes the Y117L mutant an appropriate positive control for vimentin IF disruption. Our data also demonstrate that, although S49A vimentin is capable of forming filaments (albeit at reduced levels) (Figure 3 and Figure 3—figure supplement 4), S49 is required for the characteristic *Chlamydia*-induced vimentin IF remodeling during infection (Figure 5). We believe these results indicate a specific regulatory role for the S49 glycosylation site in *Chlamydia*-induced IF remodeling. Indeed, two other lines of evidence indicate that the phenotypes we observe with the S49A mutant are due to the loss of S49 glycosylation, and not to generic loss of IF assembly. First, 5SGlcNAc treatment reduces *Chlamydia* inclusion size and extra-inclusion bacteria in cells expressing wild type vimentin, but not cells lacking vimentin (Figure 5), strongly arguing for a requirement for *both* OGT activity *and* vimentin in inclusion architecture. Second, prior studies have demonstrated that S49 is not required for IF formation, because vimentin mutants with internal head domain deletions spanning S49 are competent to form IFs in vitro and in vivo^22^. We believe these data, taken together, indicate that the IF phenotypes we observe in control and *Chlamydia*-infected cells are due to a loss of vimentin assembly regulation by O-GlcNAcylation at S49, and not due to an intrinsic, essential function of S49 per se. We have revised the manuscript in the eleventh and twelfth paragraphs of the Results section and in the second paragraph of the Discussion, to better explain these results and analyses.

7) How conserved is the S49-containing motif among other intermediate filament proteins. If conserved, are there known mutations in this residue in other intermediate filament proteins.

As the reviewer anticipates, S49 is conserved among vimentin orthologs in other vertebrates, despite the lower overall conservation of the head domain compared to the rod domain (Figure 6—figure supplement 10A)^3-8^. Among the other human type III IF family members, S49 is conserved in desmin but not peripherin or glial fibrillary acidic protein (Figure 6—figure supplement 10B). These observations suggest that control of vimentin and perhaps desmin by O-GlcNAcylation at S49 or its cognate residue (desmin S51) could be evolutionarily conserved, an important hypothesis to test in future studies. It will also be interesting to determine whether O-GlcNAcylation exerts similar effects on other IFs, since numerous IF proteins are also glycosylated in vivo^23-35^. We are not aware of characterized mutations in S51 of desmin in human disease. However, mutations in the nearby S46 residue of desmin cause myofibrillar myopathy^36,37^ and could conceivably impact on desmin glycosylation at S51 (or even S46 itself), though this idea is speculative. We have added a brief discussion of these points in the last paragraph of the Results section.

*8) Did the authors use OGA to cleave GlcNDAz* in vitro *(and see things go away) As a side note, GlcNDAz isn't turned over well in the cell and accumulates – did they note any positive/negative consequences of that in their experiments? Why cycling? Not presence of?*

An in vitro cleavage assay with recombinant-purified human OGA and GlcNDAz crosslinks is unlikely to succeed, since the covalent adducts formed by carbene insertion would almost certainly sterically occlude the crosslinked proteins from the OGA active site^38-41^. As an alternative experiment to address the reviewer’s question, we demonstrated that overexpression of human OGA in GlcNDAz-labeled cells reduces subsequent vimentin crosslink formation, whereas OGT overexpression increases it (Figure 1). As the reviewer notes, the Kohler lab has shown previously that the O-GlcNDAz moiety is turned over in cells more slowly than is the natural O-GlcNAc moiety^42^. Despite this important point, we believe our results confirm the specificity of GlcNDAz crosslinking and indicate that O-GlcNDAz cycles on vimentin, since its addition and removal can be potentiated by OGT or OGA overexpression, respectively (Figure 1). For these reasons, we previously referred to O-GlcNAc “cycling,” as opposed to “presence,” in several places in the original text. We have revised the manuscript language to remove “cycling” and to be appropriately conservative in our description of the results. Finally, we note that we have used GlcNDAz as a discovery tool for examining the biochemical effects of glycosylation on vimentin, but have not deployed GlcNDAz in any cellular functional assays (e.g., migration or *Chlamydia* infection), which might be particularly sensitive to glycan turnover kinetics. Importantly, we have not observed any obvious deleterious effects of GlcNDAz labeling (e.g., apoptosis) with the doses and incubation times used in our experiments.

9) Results, ninth paragraph: do cells in which OGT/OGA are perturbed have same characteristics? This would suggest it's cycling (esp. if it's present w/ OGA perturbation) rather than the addition of OGlcNAc (and just OGT disruption).

Our data demonstrate that GlcNDAz crosslinking in live cells is potentiated or reduced by overexpression of OGT or OGA, respectively (Figure 1). As the reviewer suggests, we believe these data suggest that O-GlcNAc is cycling on vimentin, and not merely static, since manipulating OGT or OGA levels can affect the extent of O-GlcNDAz modification and subsequent crosslinking. We have revised the manuscript to provide a more conservative description of these results. We did not observe any obvious deleterious or differential phenotypic effects of OGT and OGA overexpression with the expression levels and incubation times used in our experiments.

10) “Taken together, these results indicate that O-GlcNAcylation of vimentin, especially on S49, is required for optimal serum-stimulated cell migration.” Is that the case? If so, does it get better with TMG treatment or siRNA orf OGA?

To address the reviewer’s question, we now provide migration data with Thiamet-G-treated cells alongside vehicle- and 5SGlcNAc-treated cells (Figure 4). In cells expressing wild type vimentin, Thiamet-G treatment significantly reduced serum-induced migration, compared to vehicle alone (*p* = 0.006), albeit less dramatically than 5SGlcNAc did. In contrast, Thiamet-G treatment had no effect on serum-induced migration in cells expressing S49A (or Y117L) mutant vimentin. These results support our model that O-GlcNAcylation of vimentin at S49 is required for optimal serum-stimulated cell migration. We have revised the Results section (tenth paragraph) and Figure 4 legend to discuss these new data.

11) Results, end of eleventh paragraph: is it possible that the 5SGlcNAc influences the chlamydia itself rather than just the host cell? What if it is treated with 5SGlcNAc (even if it's not known to have OGT, it might have an effect?)

Because it is an obligate intracellular pathogen, free-living *Chlamydia* are unfortunately not culturable and therefore cannot be treated with 5SGlcNAc in isolation^43^. However, we believe it is very unlikely that 5SGlcNAc exerts its effects in our *Chlamydia* infection experiments through any pathogen-encoded target, because 5SGlcNAc has no impact on inclusion size or the number of extra-inclusion bacteria in vimentin^-/-^ host cells (Figure 5). We have revised the last paragraph of the Results section to better highlight this point.

12) Results, end of last paragraph: did increased o-glcNAc improve response to chlamydia?

To address the reviewer’s question, we now provide *Chlamydia* infection data in Thiamet-G-treated cells expressing wild type, S49A or Y117L vimentin (Figure 5—figure supplement 6). Thiamet-G had no effect on inclusion area in any of these cell types, likely because basal levels of vimentin glycosylation are relatively high (Figure 1, Figure 1—figure supplement 1 and Figure 2—figure supplement 2). Indeed, this result is consistent with our observation that 5SGlcNAc lowers GlcNDAz crosslinking of vimentin, but Thiamet-G does not raise it (Figure 1—figure supplement 1). We believe these data support our hypothesis that vimentin glycosylation is constitutively high, and that vimentin glycosylation sites and OGT activity are required for *Chlamydia*-induced IF remodeling. We discuss these new data in the tenth and eleventh paragraphs of the Results section.

13) Figure 1: were Figure 1 from the urea extraction method? This doesn't have to be chronological, but it seems important to note given that you only see it in that fraction.

We apologize that these experimental details were not provided in our initial submission. The samples analyzed in Figure 1 were prepared in a denaturing urea buffer. We have revised the legend to Figure 1 to supply this information.

14) Figure 2 would mutations of these expect to change the folding for any reason other than O-GlcNAcylation changes? There seems no independent analysis of vimentin abundance and stability.

We agree with the reviewer that the impact of O-GlcNAcylation site mutations on vimentin abundance and stability is an important question. In the revised manuscript, we provide flow cytometry data to demonstrate that vimentin-mEmerald expression is even across all reconstituted cell lines (wild type and mutants) (Figure 3—figure supplement 2). These results confirm the microscopy data from our original submission, which also reflect even expression of the wild type and mutant vimentin-mEmerald constructs across all reconstituted cell lines (Figure 3 and Figure 3—figure supplement 4). To further rule out any effect of the S49A mutation on vimentin stability, we treated cells expressing wild type or S49A vimentin with vehicle only or the proteasome inhibitor MG132 and analyzed cell lysates by IB (Figure 3—figure supplement 5). These results revealed no discernible difference between wild type and S49A vimentin in the presence or absence of proteasome inhibition. Taken together, these data indicate that the phenotypes we observe in the S49A mutant are not due to differences in vimentin expression level or stability. We have revised the eighth paragraph of the Results section to address this point.

15) Figure 3: what is being compared? WT to the other samples in the t-test or just the filaments/punctae within a sample? All samples? Statistics are unclear.

We apologize that our statistical analysis of the filament/puncta quantitation (Figure 3) was not clearly explained in the initial submission. We performed ANOVA analysis followed by pairwise t-tests to compare the filament data from each vimentin mutant to wild type, and to compare the puncta data from each vimentin mutant to wild type. All three vimentin mutants (S49A, Y117L, S49E) exhibited levels of puncta and filaments that were significantly different from those of wild type. These results support our model that S49 glycosylation regulates IF morphology. In addition, we performed t-tests comparing the puncta and filament levels of the S49A mutant and the phosphomimetic S49E mutant to each other. In this case, no significant difference was observed between the two mutants. This result (along with the lack of any literature reports of S49 phosphorylation) supports our conclusion that loss of glycosylation, and not loss of phosphorylation, likely explains the phenotypes of the S49A mutant in our assays. We have revised Figure 3, Figure 3—figure supplement 2, the Results section (eighth, ninth and tenth paragraphs) and the “Statistical analysis” subsection of the Materials and methods to clarify these points.

16) Figure 4: here it is best to argue that residues subject to glycosylation are important for cell migration. Formally, it is difficult to tease out the role of O-GlcNAc at these sites since both the GlcNAc and the backbone is altered. It’s also hard to tell what's being noted as statistically different here. I presume it's the difference between WT and the other three when serum stimulated? This should be better described either in the main text or in the legend. (i.e., vector is the -/- cell line without being reconstituted?).

We agree with the reviewer that it is challenging to stringently discriminate between the effects of lost glycosylation versus changes to the protein sequence per se in interpreting the phenotypes of individual glycosylation site mutants. As the reviewer suggests, we have revised both the Results and Discussion sections of the text to reflect the fact that our data demonstrate the importance of O-GlcNAcylation sites of vimentin for cell migration, a more parsimonious description of our results. Nevertheless, we believe two important lines of evidence suggest that it is likely loss of glycosylation, and not a change in amino acid sequence per se, that explains the phenotype of the S49A mutant in our cell migration assays (Figure 4). First, 5SGlcNAc and Thiamet-G treatment inhibit the serum-induced migration of cells expressing wild type vimentin but not cells lacking vimentin or cells expressing S49A vimentin (Figure 4). These results demonstrate that the S49 residue of vimentin is required for the inhibitory effect on cell migration of OGT or OGA inhibition. Second, prior studies have demonstrated that S49 is not required for vimentin IF assembly in vitro or in vivo, arguing against an O-GlcNAc-independent role for S49 in IF structure and function^22^. We believe the simplest interpretation of our data, taken together, is that loss of vimentin regulation by S49 glycosylation alters IF assembly state, accounting for the phenotype of the S49A mutant in the cell migration assay (Figure 4). With respect to the question of statistical comparisons, we apologize that this was not clear in the original text. For our cell migration data, we first performed an ANOVA, which compares all samples in a group at once to determine in which condition(s) the variance is significant. Then, given the significance indicated by the ANOVA, we performed pairwise t-tests for all combinations of data points. As the reviewer surmises, the serum-stimulated migration of wild type vimentin-expressing cells is significantly greater than the serum-stimulated migration of all three other cell types (vector, S49A and Y117L) (Figure 4). In addition, as noted above, 5SGlcNAc or Thiamet-G treatment significantly inhibits the serum-stimulated migration of cells expressing wild type vimentin, but not that of cells expressing any of the three mutants (Figure 4). We have revised the image and legend of Figure 4 to clarify the most important comparisons made in our statistical analyses, without unduly cluttering the graph. In addition, we have revised the Materials and methods section to explain the statistical analysis in more detail, and have now provided source data with comprehensive statistical analyses, including all pairwise t-test results, in table format (Figure 4—source data 1). To answer the final part of the reviewer’s question, “vector” refers to vimentin^-/-^ cells stably transduced with an empty version of the backbone vector used to create the wild type and mutant vimentin-mEmerald constructs. We apologize that this was unclear in our initial submission, and we have revised the legend to Figure 4 to rectify this.

17) Were the cells +5SGlcNAc serum stimulated? Are these data the same as the samples on the left just with additional datapoints of the 5SGlcNAc? Consider just condensing to one graph or noting that. If the statistics become difficult and that's why it's split out, it should be easy to note in a table or graphically when "NS" is taken out. Also, for WT serum stimulated, are there error bars?

We apologize that these details of the cell migration assay were unclear. As the reviewer anticipates, the “+5SGlcNAc” data series in Figure 4 represents cells stimulated with serum and treated with 5SGlcNAc. This also applies to the cells treated with Thiamet-G, which are new data provided in the resubmitted manuscript. We have revised the figure legend to clarify this point. Also, we have adopted the reviewer’s suggestion to simplify Figure 4 by consolidating the migration data into a single graph and by highlighting only the most important statistical comparisons. In addition, we have provided complete statistical analyses in table format as source data (Figure 4—source data 1). Finally, there are no error bars for the wild type serum-stimulated condition, because this experimental treatment was defined as maximum migration, fixed at 100% and used to normalize all other data. We have revised the legend to Figure 4 to explain this analysis more carefully.

18) Figure 5 legend: -/- is what is indicated I think?

We thank the reviewer for noticing this typo, which we have corrected in the revised manuscript. Indeed, the line should have begun “Vimentin^-/-^ HeLa cells reconstituted with WT vimentin-mEmerald…”

References

1) Yu, S.-H., Boyce, M., Wands, A.M., Bond, M.R., Bertozzi, C.R. and Kohler, J.J. Metabolic labeling enables selective photocrosslinking of O-GlcNAc-modified proteins to their binding partners. Proceedings of the National Academy of Sciences 109, 4834-4839 (2012).

2) Rodriguez, A.C., Yu, S.H., Li, B., Zegzouti, H. and Kohler, J.J. Enhanced transfer of a photocross-linking N-acetylglucosamine (GlcNAc) analog by an O-GlcNAc transferase mutant with converted substrate specificity. J Biol Chem 290, 22638-22648 (2015).

3) Lowery, J., Kuczmarski, E.R., Herrmann, H. and Goldman, R.D. Intermediate Filaments Play a Pivotal Role in Regulating Cell Architecture and Function. J Biol Chem 290, 17145-17153 (2015).

4) Herrmann, H. and Aebi, U. Intermediate Filaments: Structure and Assembly. Cold Spring Harb Perspect Biol 8(2016).

5) Chernyatina, A.A., Guzenko, D. and Strelkov, S.V. Intermediate filament structure: the bottom-up approach. Curr Opin Cell Biol 32, 65-72 (2015).

6) Koster, S., Weitz, D.A., Goldman, R.D., Aebi, U. and Herrmann, H. Intermediate filament mechanics in vitro and in the cell: from coiled coils to filaments, fibers and networks. Curr Opin Cell Biol 32, 82-91 (2015).

7) Leduc, C. and Etienne-Manneville, S. Intermediate filaments in cell migration and invasion: the unusual suspects. Curr Opin Cell Biol 32, 102-112 (2015).

8)Guharoy, M., Szabo, B., Contreras Martos, S., Kosol, S. and Tompa, P. Intrinsic structural disorder in cytoskeletal proteins. Cytoskeleton (Hoboken) 70, 550-571 (2013).

9) Ridge, K.M., Shumaker, D., Robert, A., Hookway, C., Gelfand, V.I., Janmey, P.A., Lowery, J., Guo, M., Weitz, D.A., Kuczmarski, E. and Goldman, R.D. Methods for Determining the Cellular Functions of Vimentin Intermediate Filaments. Methods Enzymol 568, 389-426 (2016).

10) Boyce, M., Carrico, I.S., Ganguli, A.S., Yu, S.-H., Hangauer, M.J., Hubbard, S.C., Kohler, J.J. and Bertozzi, C.R. Metabolic cross-talk allows labeling of O-linked β-N-acetylglucosamine-modified proteins via the N-acetylgalactosamine salvage pathway. Proc Natl Acad Sci U S A 108, 3141-3146 (2011).

11) Palaniappan, K.K., Hangauer, M.J., Smith, T.J., Smart, B.P., Pitcher, A.A., Cheng, E.H., Bertozzi, C.R. and Boyce, M. A chemical glycoproteomics platform reveals O-GlcNAcylation of mitochondrial voltage-dependent anion channel 2. Cell Rep 5, 546-552 (2013).

12) Chen, P.H., Smith, T.J., Wu, J., Siesser, P.F., Bisnett, B.J., Khan, F., Hogue, M., Soderblom, E., Tang, F., Marks, J.R., Major, M.B., Swarts, B.M., Boyce, M. and Chi, J.T. Glycosylation of KEAP1 links nutrient sensing to redox stress signaling. EMBO J 36, 2233-2250 (2017).

13) Dong, F., Su, H., Huang, Y., Zhong, Y. and Zhong, G. Cleavage of host keratin 8 by a Chlamydia-secreted protease. Infect Immun 72, 3863-3868 (2004).

14) Kumar, Y. and Valdivia, R.H. Actin and intermediate filaments stabilize the Chlamydia trachomatis vacuole by forming dynamic structural scaffolds. Cell Host Microbe 4, 159-169 (2008).

15) Jorgensen, I., Bednar, M.M., Amin, V., Davis, B.K., Ting, J.P., McCafferty, D.G. and Valdivia, R.H. The Chlamydia protease CPAF regulates host and bacterial proteins to maintain pathogen vacuole integrity and promote virulence. Cell Host Microbe 10, 21-32 (2011).

16) Snavely, E.A., Kokes, M., Dunn, J.D., Saka, H.A., Nguyen, B.D., Bastidas, R.J., McCafferty, D.G. and Valdivia, R.H. Reassessing the role of the secreted protease CPAF in Chlamydia trachomatis infection through genetic approaches. Pathog Dis 71, 336-351 (2014).

17) Bednar, M.M., Jorgensen, I., Valdivia, R.H. and McCafferty, D.G. Chlamydia protease-like activity factor (CPAF): characterization of proteolysis activity in vitro and development of a nanomolar affinity CPAF zymogen-derived inhibitor. Biochemistry 50, 7441-7443 (2011).

18) Meier, M., Padilla, G.P., Herrmann, H., Wedig, T., Hergt, M., Patel, T.R., Stetefeld, J., Aebi, U. and Burkhard, P. Vimentin coil 1A-A molecular switch involved in the initiation of filament elongation. J Mol Biol 390, 245-261 (2009).

19) Helfand, B.T., Mendez, M.G., Murthy, S.N., Shumaker, D.K., Grin, B., Mahammad, S., Aebi, U., Wedig, T., Wu, Y.I., Hahn, K.M., Inagaki, M., Herrmann, H. and Goldman, R.D. Vimentin organization modulates the formation of lamellipodia. Mol Biol Cell 22, 1274-1289 (2011).

20) Robert, A., Rossow, M.J., Hookway, C., Adam, S.A. and Gelfand, V.I. Vimentin filament precursors exchange subunits in an ATP-dependent manner. Proc Natl Acad Sci U S A 112, E3505-3514 (2015).

21) Jiu, Y., Peranen, J., Schaible, N., Cheng, F., Eriksson, J.E., Krishnan, R. and Lappalainen, P. Vimentin intermediate filaments control actin stress fiber assembly through GEF-H1 and RhoA. J Cell Sci 130, 892-902 (2017).

22) Shoeman, R.L., Hartig, R., Berthel, M. and Traub, P. Deletion mutagenesis of the amino-terminal head domain of vimentin reveals dispensability of large internal regions for intermediate filament assembly and stability. Exp Cell Res 279, 344-353 (2002).

23) King, I.A. and Hounsell, E.F. Cytokeratin 13 contains O-glycosidically linked N-acetylglucosamine residues. J Biol Chem 264, 14022-14028 (1989).

24) Chou, C.F., Smith, A.J. and Omary, M.B. Characterization and dynamics of O-linked glycosylation of human cytokeratin 8 and 18. J Biol Chem 267, 3901-3906 (1992).

25) Ku, N.O., Toivola, D.M., Strnad, P. and Omary, M.B. Cytoskeletal keratin glycosylation protects epithelial tissue from injury. Nat Cell Biol 12, 876-885 (2010).

26) Dong, D.L., Xu, Z.S., Hart, G.W. and Cleveland, D.W. Cytoplasmic O-GlcNAc modification of the head domain and the KSP repeat motif of the neurofilament protein neurofilament-H. J Biol Chem 271, 20845-20852 (1996).

27) Dong, D.L., Xu, Z.S., Chevrier, M.R., Cotter, R.J., Cleveland, D.W. and Hart, G.W. Glycosylation of mammalian neurofilaments. Localization of multiple O-linked N-acetylglucosamine moieties on neurofilament polypeptides L and M. J Biol Chem 268, 16679-16687 (1993).

28) Ludemann, N., Clement, A., Hans, V.H., Leschik, J., Behl, C. and Brandt, R. O-glycosylation of the tail domain of neurofilament protein M in human neurons and in spinal cord tissue of a rat model of amyotrophic lateral sclerosis (ALS). J Biol Chem 280, 31648-31658 (2005).

29) Deng, Y., Li, B., Liu, F., Iqbal, K., Grundke-Iqbal, I., Brandt, R. and Gong, C.X. Regulation between O-GlcNAcylation and phosphorylation of neurofilament-M and their dysregulation in Alzheimer disease. Faseb J 22, 138-145 (2008).

30) Cheung, W.D. and Hart, G.W. AMP-activated protein kinase and p38 MAPK activate O-GlcNAcylation of neuronal proteins during glucose deprivation. J Biol Chem 283, 13009-13020 (2008).

31) Slawson, C., Lakshmanan, T., Knapp, S. and Hart, G.W. A mitotic GlcNAcylation/phosphorylation signaling complex alters the posttranslational state of the cytoskeletal protein vimentin. Mol Biol Cell 19, 4130-4140 (2008).

32) Wang, Z., Pandey, A. and Hart, G.W. Dynamic interplay between O-linked N-acetylglucosaminylation and glycogen synthase kinase-3-dependent phosphorylation. Mol Cell Proteomics 6, 1365-1379 (2007).

33) Srikanth, B., Vaidya, M.M. and Kalraiya, R.D. O-GlcNAcylation determines the solubility, filament organization, and stability of keratins 8 and 18. J Biol Chem 285, 34062-34071 (2010).

34) Kakade, P.S., Budnar, S., Kalraiya, R.D. and Vaidya, M.M. Functional Implications of O-GlcNAcylation-dependent Phosphorylation at a Proximal Site on Keratin 18. J Biol Chem 291, 12003-12013 (2016).

35) Tao, G.Z., Kirby, C., Whelan, S.A., Rossi, F., Bi, X., MacLaren, M., Gentalen, E., O'Neill, R.A., Hart, G.W. and Omary, M.B. Reciprocal keratin 18 Ser48 O-GlcNAcylation and Ser52 phosphorylation using peptide analysis. Biochem Biophys Res Commun 351, 708-712 (2006).

36) Baker, L.K., Gillis, D.C., Sharma, S., Ambrus, A., Herrmann, H. and Conover, G.M. Nebulin binding impedes mutant desmin filament assembly. Mol Biol Cell 24, 1918-1932 (2013).

37) Selcen, D., Ohno, K. and Engel, A.G. Myofibrillar myopathy: clinical, morphological and genetic studies in 63 patients. Brain 127, 439-451 (2004).

38) Li, B., Li, H., Hu, C.W. and Jiang, J. Structural insights into the substrate binding adaptability and specificity of human O-GlcNAcase. Nat Commun 8, 666 (2017).

39) Li, B., Li, H., Lu, L. and Jiang, J. Structures of human O-GlcNAcase and its complexes reveal a new substrate recognition mode. Nat Struct Mol Biol 24, 362-369 (2017).

40) Roth, C., Chan, S., Offen, W.A., Hemsworth, G.R., Willems, L.I., King, D.T., Varghese, V., Britton, R., Vocadlo, D.J. and Davies, G.J. Structural and functional insight into human O-GlcNAcase. Nat Chem Biol 13, 610-612 (2017).

41) Elsen, N.L., Patel, S.B., Ford, R.E., Hall, D.L., Hess, F., Kandula, H., Kornienko, M., Reid, J., Selnick, H., Shipman, J.M., Sharma, S., Lumb, K.J., Soisson, S.M. and Klein, D.J. Insights into activity and inhibition from the crystal structure of human O-GlcNAcase. Nat Chem Biol 13, 613-615 (2017).

42) Rodriguez, A.C. and Kohler, J.J. Recognition of diazirine-modified O-GlcNAc by human O-GlcNAcase. Medchemcomm 5, 1227-1234 (2014).

43) Campbell, L.A. and Kuo, C.C. Cultivation and laboratory maintenance of Chlamydia pneumoniae. Curr Protoc Microbiol Chapter 11, Unit11B 11 (2009).

44) Mellacheruvu, D., Wright, Z., Couzens, A.L., Lambert, J.P., St-Denis, N.A., Li, T., Miteva, Y.V., Hauri, S., Sardiu, M.E., Low, T.Y., Halim, V.A., Bagshaw, R.D., Hubner, N.C., Al-Hakim, A., Bouchard, A., Faubert, D., Fermin, D., Dunham, W.H., Goudreault, M., Lin, Z.Y., Badillo, B.G., Pawson, T., Durocher, D., Coulombe, B., Aebersold, R., Superti-Furga, G., Colinge, J., Heck, A.J., Choi, H., Gstaiger, M., Mohammed, S., Cristea, I.M., Bennett, K.L., Washburn, M.P., Raught, B., Ewing, R.M., Gingras, A.C. and Nesvizhskii, A.I. The CRAPome: a contaminant repository for affinity purification-mass spectrometry data. Nat Methods 10, 730-736 (2013).